# Doxorubicin-induced p53 interferes with mitophagy in cardiac fibroblasts

**T. R. Mancilla[1,2], L. R. Davis[2], G. J. Aune** [ID][2,3]*

**1** Department of Cellular and Integrative Physiology, University of Texas Health Science Center San Antonio, San Antonio, TX, United States of America, **2** Greehey Children's Cancer Research Institute, University of Texas Health Science Center San Antonio, San Antonio, TX, United States of America, **3** Department of Pediatrics, Division of Hematology-Oncology, University of Texas Health Science Center San Antonio, San Antonio, TX, United States of America

* Aune@uthscsa.edu

## Abstract

Anthracyclines are the critical component in a majority of pediatric chemotherapy regimens due to their broad anticancer efficacy. Unfortunately, the vast majority of long-term child-hood cancer survivors will develop a chronic health condition caused by their successful treatments and severe cardiac disease is a common life-threatening outcome that is unequivocally linked to previous anthracycline exposure. The intricacies of how anthracy-clines such as doxorubicin, damage the heart and initiate a disease process that progresses over multiple decades is not fully understood. One area left largely unstudied is the role of the cardiac fibroblast, a key cell type in cardiac maturation and injury response. In this study, we demonstrate the effect of doxorubicin on cardiac fibroblast function in the presence and absence of the critical DNA damage response protein p53. In wildtype cardiac fibroblasts, doxorubicin-induced damage correlated with decreased proliferation and migration, cell cycle arrest, and a dilated cardiomyopathy gene expression profile. Interestingly, these doxorubicin-induced changes were completely or partially restored in p53$^{-/-}$ cardiac fibroblasts. Moreover, in wildtype cardiac fibroblasts, doxorubicin produced DNA damage and mitochondrial dysfunction, both of which are well-characterized cell stress responses induced by cytotoxic chemotherapy and varied forms of heart injury. A 3-fold increase in p53 (p = 0.004) prevented the completion of mitophagy (p = 0.032) through sequestration of Parkin. Interactions between p53 and Parkin increased in doxorubicin-treated cardiac fibroblasts (p = 0.0003). Finally, Parkin was unable to localize to the mitochondria in wildtype cardiac fibroblasts, but mitochondrial localization was restored in p53$^{-/-}$ cardiac fibroblasts. These findings strongly suggest that cardiac fibroblasts are an important myocardial cell type that merits further study in the context of doxorubicin treatment. A more robust knowl-edge of the role cardiac fibroblasts play in the development of doxorubicin-induced cardio-toxicity will lead to novel clinical strategies that will improve the quality of life of cancer survivors.

**Data Availability Statement:** All relevant data are within the manuscript and its Supporting Information files. Data for gene expression as well as all original western blots have been uploaded

with this submission and are included in a file named: 'S1_raw_images'.

**Funding:** GJA was supported by the following: • St. Baldrick's Foundation Scholar (Career Development Award) https://www.stbaldricks.org • Turn it Gold Foundation https://turnitgold.org TRM was supported by the following: • NIH T32GM113896 / STXMSTP https://www.nigms.nih.gov/ The funders had no role in study design, data collection and analysis, decision to publish, or preparation of the manuscript.

**Competing interests:** NO authors have competing interest.

# Introduction

Due to improved treatment regimens, by 2020 there will be over 500,000 childhood cancer survivors in the U.S. [1] Unfortunately, by age 50 over half of these long-term survivors will suffer from at least one debilitating or even fatal chronic health condition [2]. The most common fatal late complications in these survivors are secondary neoplasms and heart disease [3]. Epidemiological data has unequivocally linked doxorubicin (DOX), a chemotherapy agent used in over 50% of pediatric cancer cases [4], to acute and chronic cardiotoxicity [5–7]. While the drug has been in use since the 1970's, clinical and preclinical research has yet to produce an effective strategy to prevent chronic cardiotoxicity before or after exposure.

DOX toxicity manifests differently in cancer survivors treated as adults versus those treated in childhood [8, 9]. A barrier to understanding the pathologic mechanisms of chronic DOX cardiotoxicity in survivors of pediatric cancer is the long latent period between exposure and development of cardiac disease. In patients that are treated during childhood, cardiac dysfunction is often undetectable for two or more decades. While childhood cancer survivors suffer from chronic cardiotoxicity, acute toxicity has essentially been eliminated in this population due a strict adherence to lifetime cumulative dose limits [10, 11]. Molecular and physiological differences also contribute to the divergence in pediatric and adult patient outcomes. At the time of DOX exposure, children are undergoing cardiac growth and maturation [12]. Dysfunction of the various cells of the heart due to DOX exposure could lead to aberrant growth and maturation. (CFs, in particular, use paracrine signaling to mediate cardiac myocyte hypertrophy [13–15] and ECM remodeling, [16, 17] which are both necessary components of intact cardiac growth and maturation.

CFs are key players in the myocardial stress response [18–24]. In the maturing heart, CFs regulate cardiac myocyte growth [25, 26]. Throughout life, CFs mediate injury response in the heart. CF dysfunction in a maturing heart could cause a progressive and sustained pathology. One critical regulator of CF function are healthy mitochondria. The effect of DOX on CF mitochondrial health has not been previously investigated. DOX induces mitochondrial dysfunction and depolarization in cardiac myocytes. Mitophagy, the degradation and recycling of mitochondria, is one cellular process that maintains mitochondrial quality control. Without the ability to undergo mitophagy, a cell accumulates dysfunctional and unhealthy mitochondria that perpetuate the enhanced production of damaging reactive oxygen species (ROS) [27, 28]. The cellular and myocardial consequences of DOX-induced stress in CFs, such as mitochondrial dysfunction, are unknown. However, it is plausible that over time, an accumulation of damaged mitochondria could create a cardiac milieu primed for adverse remodeling and gradual progression to overt heart failure.

Oxidative stress in fibroblasts has been investigated in other tissue types, predominantly skin fibroblasts. One study showed that excess ROS upregulated matrix metalloproteinase (MMP) expression and downregulated collagen expression [29], altering the extracellular matrix (ECM). Mitochondrial dysfunction in skin fibroblasts leads to decreased proliferation and ATP production, and in one case depolarization of the mitochondrial membrane [30, 31].

Despite an abundance of current and past research, a protective intervention has not yet been developed and proven effective in limiting chronic DOX-induced cardiotoxicity. Some preclinical evidence points to exercise as a mitigating intervention for DOX-induced cardiotoxicity, but the underlying mechanisms are not clearly understood [32]. In addition, more sophisticated preclinical modalities are needed to measure heart dysfunction, so that preclinical research is more closely aligned with clinical reality [33]. The CF, with its role in cardiac maturation and injury response, is a logical focus of research. In this study we examine DOX-induced inhibition of mitophagy in CFs that leads to inadequate stress responses and cellular

dysfunction. Through this work we sought to establish DOX's effect on the CF as a foundation for further research focused on the role CFs may play in the overall progression of DOX-induced cardiac disease.

## Material & methods

### Animal care & protocol

C57BL/6J mice were purchased from The Jackson Laboratory (Bar Harbor, ME) and p53$^{+/-}$ mice were a gift from the laboratory of Guillermina Lozano at UT MD Anderson Cancer Center (Houston, TX). Animals were bred in-house to maintain the C57BL/6J (WT) colony and to develop p53$^{-/-}$ mice for CF isolation. Mice housing rooms were equipped with temperature control and a 12-hour light-dark cycle. Mice were allowed water and standard chow *ad libitum*. Offspring were weaned at three weeks of age and separated according to sex. All mouse procedures were carried out in agreement with the Guide for the Care and Use of Laboratory Animals (National Research Council, National Academy Press, Washington DC, USA 2011) and were approved by the Institutional Animal Care and Use Committee at the University of Texas Health at San Antonio.

### Genotyping

The p53$^{-/-}$ genotype was determined with the REDExtract-N-AMP Tissue PCR Kit (Sigma). DNA was amplified with the following primers: WT sense 5'-AGGCTTAGAGGTGCAAG CTG-3', mutant sense 5'-CAGCCTCTGTCCCACATACACT-3', and common antisense: 5'-TGGATGGTGGTATACTCAGAGC-3'. The expected weight for the wildtype gene is 321 bp and 110 bp for the mutant gene. Samples were run on a 1.5% agarose gel and expected band weights are 321 bp (WT) and 100 bp (p53 mutated).

### Cell isolation, culture, and treatment

CFs were isolated from the left ventricle of 6- to 8-week-old WT and p53$^{-/-}$ mice. The main vessels, atria, and right ventricle were removed from excised hearts. Tissue was dissociated by mechanical disruption and collagenase II (Worthington) incubations. Cells were expanded in DMEM/F-12 50/50 media (Corning) with 10% fetal bovine serum and 1% penicillin streptomycin solution. For studies, CFs were plated at ~80% confluence and adhered overnight. CFs were exposed for three hours to 1, 3, or 5 μM DOX (TEVA) or 3 μM FCCP (Abcam). After treatment, cells were washed with HBSS. Assays were performed 3–72 hours after treatment. "Standard treatment" indicates that cells were treated with 3μ DOX for 3 hrs. Variations in experimental format are noted where relevant in the methods below.

### Functional studies

For proliferation studies, cells were plated at ~30% confluence and images obtained every 2 hours for 72 hours. An IncuCyte® Live-Cell Analysis System (Essen Biosciences) collected images and proprietary IncuCyte® software calculated confluence at each time point.

Migration assays were performed with the QCM Chemotaxis Cell Migration Assay (EMD Millipore). Cells were treated with DOX or saline and after 24 hours transferred to a Boyden chamber membrane (8 μm pore) in serum-free media. The lower chamber was filled with 10% FBS media. After 24 hours, migrated cells were stained and the absorbance measured. (EMD Millipore).

Cell cycle arrest was assessed by propidium iodide (PI) DNA-staining. Three hours after treatment, cells were detached, and a single cell suspension was generated. Fluorescence was quantified by the UTHSCSA Flow Cytometry Core Center.

## Viability assays

Cell mass after 24 hours was determined with a sulforhodamine B (SRB) protein dye assay. Cells were treated, fixed, and protein was stained with SRB. Absorbance at 595 nm was compared to time zero fixed cells. Membrane integrity at 3 and 24 hours after treatment was evaluated with a Trypan Blue Exclusion assay. Cells were incubated with trypan blue and the proportion of clear cells to total cells was calculated. Cellular metabolism was tested via the reduction of MTT. Cells were incubated in MTT solution and formazan crystals were solubilized with DMSO. Absorbance was read at 540 nm.

## Gene expression profiling

Total RNA was collected from CFs 24 hours after treatment of 3 μM DOX, utilizing the RNeasy Mini Kit (Qiagen). Samples were prepared for real tme qPCR with the RT$^2$ First Strand Kit and the RT$^2$ SYBR Green qPCR Mastermix (Qiagen). Two pathway-directed RT$^2$ Profiler PCR Arrays were selected from Qiagen: Mouse Extracellular Matrix & Adhesion Molecules and Mouse Cytokines & Chemokines.

## Markers of mitochondrial health

ROS production and mitochondrial mass MitoSOX Red and Mitotracker Green fluorescent dyes (Fisher) were used to evaluate ROS production and mitochondrial mass with microscopy and flow cytometry. After treatment, cells were incubated in the dyes in. A single cell suspension was generated and samples were run by the UTHSCSA Flow Cytometry Core. Membrane potential was assessed with a JC-1 Mitochondrial Membrane Potential Assay Kit (Abcam). Before treatment, cells were stained with the JC-1 dye. Aggregate and monomer concentrations were measured 3 hours after the treatment with the following wavelengths: ex/em 535/590 nm and ex/em 470/530 nm, respectively.

## Cellular ROS production

Cellular ROS production was measured with the DCFDA (2,7-dichlrorfluorescin diacetate) Cellular ROS Detection Assay Kit (Abcam) according to kit instructions. Cells were stained with 25 μM DCFDA for 45 minutes. After washing, the cells were treated with 1, 5, or 10 μM DOX. Tertbutyl hydrogen peroxide (TBHP) treated cells and DOX-treated cells without stain acted as positive and negative controls, respectively.

## Seahorse energy phenotype

Mitochondrial function was assessed with the Seahorse Energy Phenotype Assay (Agilent Technologies) according to the kit protocol. Briefly, cells were plated on a Seahorse XFp 96-well plate. Cells were treated with 1, 3, or 5 μM DOX for three hours then washed with Seahorse Assay Media (10 mM glucose, 1 mM pyruvate, and 2 mM L-Glutamine) and placed in a non-$CO_2$ incubator for one hour. A stock solution of 10 μM oligomycin and 10 μM FCCP was prepared for automatic administration during the assay. The Seahorse assay software obtained measurements of oxygen consumption and acidification levels before and after the addition of the mitochondrial stressors, oligomycin and FCCP. Optimal cell density and FCCP and oligomycin concentrations were predetermined from a dose response titration of the three variables. Values were determined from Agilent recommended baseline levels of oxygen consumption and changes incited by stressor addition.

## Immunoblotting

For localization studies, cellular compartments were isolated using the Cell Fractionation Kit (Abcam). Fractionation was verified with GAPDH (EMD Millipore) for cytosolic fractions, nuclear lamin for nuclear fractions, and VDAC for mitochondrial fractions (Abcam. Protein content was measured with Pierce BCA Protein Assay (ThermoFisher Scientific). Samples were separated by electrophoresis and transferred to a nitrocellulose membrane. Li-Cor fluorescent-conjugated antibodies were used for visualization on a Li-Cor Clx imaging system. Band densitometry was calculated using Image J software (NIH) and standardized with total protein values (REVERT Total Protein Stain, Li-Cor) from each lane.

## Immunofluorescent cytometry

After treatment, cells were fixed in 4% paraformaldehyde, permeabolized, and blocked in 5% bovine serum albumin. Cells were incubated in primary antibody overnight, washed, incubated in fluorescent secondary antibodies from Abcam, and mounted with Prolong Gold Antifade reagent.

## Mitophagy assay

Samples were prepared with the Dojindo Molecular Technologies, Inc Mitophagy Detection Kit according to manufacturer's instructions. Briefly, cells were incubated in a mitochondrial-binding dye prior to treatment. Three hours after treatment ceased, cells were incubated with a lysosome dye. Samples were analyzed and cell populations were quantified under the supervision of the UTHSCSA Flow Cytometry Core.

## Proximity ligation assay

Cells were plated on glass coverslips, treated with DOX, and washed. Three hours after treatment, cells were fixed in 4% paraformaldehyde and permeabilized with Triton-X. The proximity ligation assay was performed using the Duolink In Situ Red Starter Kit Mouse/Rabbit (Sigma-Aldrich). After blocking, cells were incubated overnight at 4˚C in a mixture of p53 and Parkin antibodies. Negative controls included replicates that were not incubated in the p53 antibody, the Parkin antibody, or neither antibody.

Cells were incubated in the following solutions sequentially, PLUS anti-rabbit and MINUS anti-mouse probes, a ligation solution, and an amplification buffer. Cells were mounted with a DAPI-containing solution. At least ten fields of view per sample were obtained. Fluorescent signal was quantified using Image J and normalized to nuclear signal.

## Statistical analysis

All statistical tests were performed using GraphPad Prism 7. A student's T-test was used for comparisons between two groups, with a Welch's correction for unequal variance when necessary. Comparisons between more than two groups were analyzed with a one-way ANOVA. A second order polynomial regression was fit to growth curves and comparisons of fit were conducted. Cell cycle distribution was analyzed with a $Chi^2$ test. The threshold for the p value was set at 0.05, except for the gene array profiles where a more stringent threshold of 0.01 was used. Graphs are representative of at least 3 biological replicates and presented as average ± SEM.

# Results

## Measures of cardiac function

CFs respond to injury by proliferating in and migrating to areas of damage. The ability to carry out these two functions are essential to a CFs' role in maturation and injury response.

Therefore, the first items assessed when determining the effects of DOX on CFs were growth and migration. Primary CFs, isolated from C57BL/6J and p53$^{-/-}$ mice were treated for 3 hours with 3.0 μM DOX, the drug washed away, and cells were placed in an IncuCyte® for live cell imaging.

Fig 1A and 1B shows DOX significantly reduced growth (p<0.0001) in both cell strains. By 72 hours, the confluence of the treated cells was almost half that of untreated cells. After 36 hours, it appears that the reduction in confluence is due more to cell death than decreased proliferation. Fig 1C demonstrates the mask applied to measure cell confluence.

Migration was assessed by cell movement through a membrane in a modified Boyden Chamber. DOX impeded migration by ~15% in WT cells (p<0.0001), but p53 deletion restored the cell's migratory ability (Fig 1D). After treatment and PI incubation, cell cycle distribution was assessed via flow cytometry. WT cells showed a significant difference between the cell cycle distributions (p = 0.001), most likely indicating arrest at the p53-dependent G1/S checkpoint. There was no change in the percentage of cells in the G2 phase, with the complimentary increase, decrease seen in the G1 and S phases, respectively. P53$^{-/-}$ did not show the same cell cycle arrest (Fig 1G–1J). The reduction in proliferation therefore has a different etiology in p53$^{-/-}$ cells.

## Cell viability after DOX exposure

To determine if the reduced growth and migration were due to a decrease in proliferation, an increase in cell death, or some combination, viability assays evaluating membrane integrity, protein mass, and cellular metabolism were performed. Neither WT nor p53$^{-/-}$ cells showed changes in cell viability at doses of 1, 3, or 5 μM at 3 or 24 hrs post-treatment. In Fig 1E and 1F, it can be seen that 95% of cells were viable in all groups. SRB was used to stain protein content in cells 24 hours after treatment with 3 μM DOX (Fig 1H and 1I). A 2-fold decrease in protein content was seen after 24 hours in the WT treated cells (p<0.0001). The protein assay indicated a slight increase in protein production after DOX-exposure compared in p53$^{-/-}$ cells (p = 0.008).

Reduction of the substrate MTT by oxidoreductase enzymes is an indication of cellular metabolic activity. MTT reduction remained constant compared to control at 3 hours post-treatment in all three doses. 24 hours after treatment, metabolic activity decreased in a dose-dependent fashion (Fig 1K) in WT cells. The decreases at 3 and 5 μM were significant compared to both WT control and 1 μM DOX. For p53$^{-/-}$ cells the reduction seen in the 24-hour 5 μM group was statistically significant compared to the control and 1 μM groups, while the 3 μM group was only significant compared to control- a slight attenuation from the WT studies (Fig 1L).

The decreased ability in WT cells to proliferate and migrate would have an adverse effect on the ability of a CF to perform its duties in response to injury in the heart. While WT cells remain viable after DOX exposure, their decreased protein content and metabolism could indicate an inability to mount a sufficient stress or injury response. WT cells exposed to DOX demonstrated an overall reduction in the ability to perform important CF functions. While unable to completely ameliorate the changes induced by DOX exposure, the p53$^{-/-}$ cells were able to maintain some of the functions necessary for a CF to respond to injury in the heart.

## Gene expression profile

To determine how components of ECM remodeling and inflammatory signaling were affected by DOX, pathway-directed gene arrays were used to assess the expression of 168 relevant genes. mRNA was isolated 24 hours after standard treatment. Expression was measured using real time reverse transcriptase polymerase chain reaction (RT$^2$-PCR).

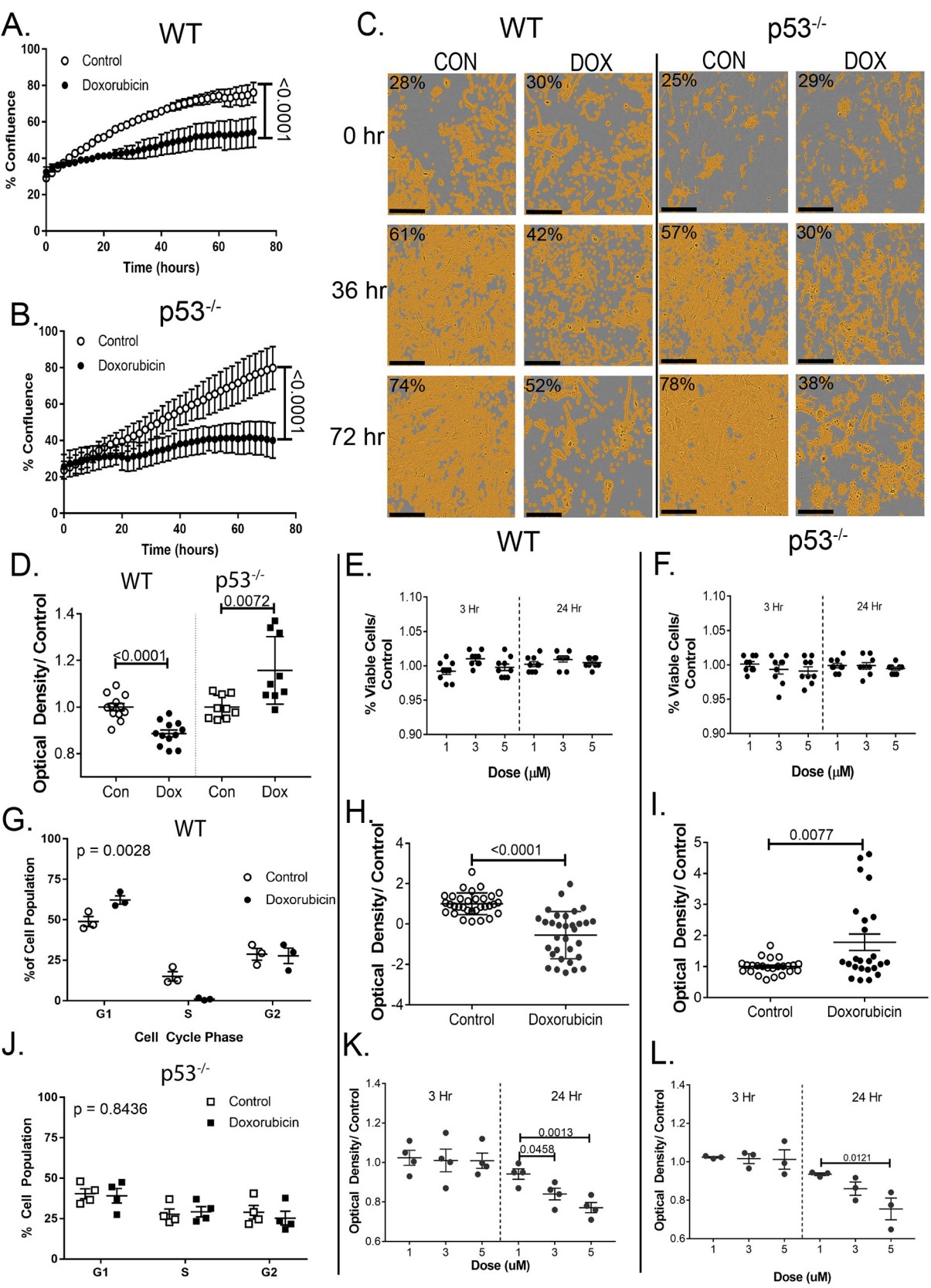

**Fig 1. DOX alters cardiac fibroblast function.** Growth curves over 72 hours of (A) WT and (B) p53[-/-] after standard DOX exposure. (C) demonstrates the mask used to measure confluence. (D) Twenty-four hours after standard DOX exposure cells were plated on a modified Boyden chamber to assess migration. (E), (F) trypan blue exclusion was used to assess cell viability. (G), (J) Relative genetic content was assessed with PI dye 3 hrs after standard treatment. Samples were fixed and sorted via flow cytometry. (H), (I) Three hrs after standard treatment, cellular protein content was labeled with SRB dye and absorbance was measured to assess relative quantity. (K), (L) Changes from baseline metabolic activity were assessed with an MTT assay. Cell cycle distribution was analyzed with a Chi$^2$ test. Graphs are an average of 3 biological replicates (with 8 technical replicates each) ± SEM. Scale bar (C) is 200 μm.

Of 168 genes, 50 were differentially expressed, a 2-fold increase or decrease between the control and treated WT cells (**Table 1**). Twenty-eight genes were upregulated and 22 were downregulated in the treated WT cells. Structural genes, such as Col1a1, Col4a2, Col5a1, and Col6a1 were downregulated, while effectors of remodeling, such as matrix metalloproteinases (MMP) 3, 8, and 12, were upregulated. Additionally, the MMP inhibitor, TIMP3, was downregulated. Seven of the 10 adhesion molecules with differential expression were downregulated in the DOX-treated WT cells. Sixteen inflammatory genes were significantly expressed, and all but two, were upregulated. Ccl2 and its receptor, Ccr4, were increased 40- and 50-fold respectively. Markers of cardiac dysfunction, Cxcl10 and Cxcl11, were increased 26- and 20-fold compared to WT control.

Only 13 genes were differentially regulated between the control and treated p53[-/-] cells (**Table 2**). Two genes, Hapln1 and Cdh3, were down-regulated, while the other 11 were upregulated. Most notable was the attenuation of the inflammatory genes. Only seven cytokines showed increased expression, and by 4-fold or less. Ccr4, which was increased 50-fold in the WT cells, was only increased 3-fold in the p53[-/-] cells. Ccl2, Cxcl10, and Cxcl11 expression were not significantly increased. Only three ECM remodeling genes were up-regulated and only one ECM structural gene was down-regulated.

After exposure to DOX, gene expression of a number of pro-inflammatory chemokines is upregulated, and in some cases upregulated quite markedly. CF chemokine upregulation in the heart is a part of the injury response role that CFs play. However, it is known that CF injury response and related cardiac remodeling rely on a regulatory balance that once tipped can lead to the development of adverse cardiac remodeling. The p53[-/-] strain of cells did not respond in

**Table 1. Significant WT gene profiling changes.**

| ECM Structural | | ECM Remodeling | | Inflammatory Cytokines | | | | Adhesion Molecules | |
|---|---|---|---|---|---|---|---|---|---|
| Gene | Fold Change | Gene | Fold Change | Gene | Fold Change | Gene | Fold Change | Gene | Fold Change |
| Lamb3 | 5.4 | Mmp15 | 24.4 | Ccr4 | 53.0 | Csf1 | 7.0 | Cdh2 | 12.2 |
| Ecm1 | 2.3 | Mmp10 | 12.4 | Ccl2 | 42.5 | Ccl9 | 4.3 | Icam1 | 2.9 |
| Lama1 | -7.7 | Mmp8 | 7.5 | Cxcl10 | 26.4 | Il33 | 4.0 | Vcam1 | 2.5 |
| Col5a1 | -3.5 | Mmp13 | 4.6 | Cxcl11 | 20.5 | Il17b | 3.6 | Itgal | -3.7 |
| Col4a2 | -3.1 | Mmp12 | 3.4 | Osm | 18.1 | Nampt | 3.3 | Pcam1 | -3.0 |
| Lamc | -3.1 | Adamts8 | 3.3 | Ccl3 | 14.4 | Bmp2 | 3.0 | Ctnna2 | -3.0 |
| Emilin1 | -3.0 | Mmp3 | 3.0 | CD40l | 8.9 | Il4 | 3.0 | Ncam1 | -2.7 |
| Col6a1 | -2.9 | Adamts5 | -4.4 | Ccl24 | 8.4 | Cxcl5 | 2.1 | Itgax | -2.6 |
| Lama2 | -2.8 | Adamts1 | -3.6 | Il11 | -3.4 | Ccr8 | -2.1 | Itga4 | -2.3 |
| Col1a1 | -2.6 | Spp1 | -2.3 | | | | | Itgam | -2.1 |
| Fn1 | -2.3 | Timp3 | -2.2 | | | | | | |

168 genes were measured after standard treatment, along with controls for genomic DNA contamination, polymerase activity, and normalization. Significant gene changes were those that passed a limit of detection threshold, were two-fold greater or lesser than the control, and the p-value was <0.01. Red indicates an up-regulation in expression compared to control, while blue indicates a down-regulation. n = 6.

**Table 2. Gene changes in p53$^{-/-}$ cells exposed to doxorubicin.**

| ECM Structural | | ECM Remodeling | | Inflammatory Cytokines | | | | Adhesion Molecules | |
|---|---|---|---|---|---|---|---|---|---|
| Gene | Fold Change | Gene | Fold Change | Gene | Fold Change | Gene | Fold Change | Gene | Fold Change |
| Lama3 | 8.3 | Adamts8 | 2.91 | Hc | 4.08 | Cxcl10 | 2.67 | Cdh3 | -2.8 |
| Hapln1 | -2.42 | Mmp13 | 2.87 | Il17a | 3.23 | Osm | 2.36 | | |
| | | Mmp10 | 2.45 | Il17b | 3.12 | Ccl3 | 2.06 | | |
| | | | | Ccr4 | 3.11 | | | | |

168 genes were measured after standard treatment, along with controls for genomic DNA contamination, polymerase activity, and normalization. Significant gene changes were those that passed a limit of detection threshold, were two-fold greater or lesser than the control, and the p-value was <0.01.Red cells indicate an up-regulation in expression compared to control, while blue cells indicate a down-regulation. n = 6.

the same manner. A much more mild upregulation of pro-inflammatory cytokines was demonstrated, along with a very different response in the ECM genes. The variations in the ECM genes that were differentially expressed in each strain suggests a differential capacity for cardiac remodeling in each strain.

## Doxorubicin-induced oxidative stress & mitochondrial health

In every cell where it has been studied, DOX induces the production of ROS. We determined that this was true in CFs as well. Production of ROS was assessed by prestaining primary CFs with a dye that fluoresces once oxidized. During treatment with DOX, ROS production increased in the cells in a dose-dependent manner (**Fig 2A**). DOX induction of ROS can occur with mitochondrial enzymes, such as NADH and NADPH. DOX can also elicit a chain reaction by interacting with iron to create free radicals. This can happen anywhere in the cell where there is $Fe^{2+}$ or $Fe^{3+}$, including the cytoplasm [34, 35]. The increase in cellular ROS is a combination of multiple mechanisms.

To view mitochondrial-specific ROS production, cells were plated on microscope slides and stained with MitoSOX Red after treatment. MitoSOX Red binds to mitochondria and fluoresces when oxidized. This fluorescence was quantified using flow cytometry. **Fig 2C** shows a representative staining of MitoSOX Red in control and treated cells, along with quantification of relative mitochondrial ROS (**Fig 2B**). DOX did increase the presence of ROS at the WT mitochondria compared to control (p = 0.006). This was not true for p53$^{-/-}$ cells. There is greater variability of in the production of ROS associated with mitochondria in the p53$^{-/-}$ cells whereas it is more uniformly increased in the WT cells. The variability of mitochondrial ROS production could reflect the process of mitophagy, as dysfunctional ROS-producing mitochondria are degraded. Additionally, both groups are relative to their control baseline. Cells without p53 could have a lower baseline amount of ROS creating a greater percent change in mitochondria that are producing ROS.

Mitochondrial health after DOX exposure was assessed with two fluorescent markers, Mitotracker Green and JC-1. Mitochondrial mass in primary CFs was assessed qualitatively by microscopy and quantitatively by flow cytometry. Mitochondrial mass in neither WT nor p53$^{-/-}$ CFs was changed with DOX exposure (**Fig 2F and 2G**). JC-1 dye aggregates in the polarized mitochondrial membrane. As the membrane depolarizes, the dye diffuses into the cytoplasm as monomers. Membrane potential in primary CFs was visualized with fluorescent microscopy. The fluorescent signal was quantified on a spectrophotometer. DOX decreased the average mitochondrial membrane potential, indicating depolarization of the membrane and a lower ratio of healthy to unhealthy mitochondria compared to control samples (**Fig 2D and 2E**). The change was abrogated in the p53$^{-/-}$ cells.

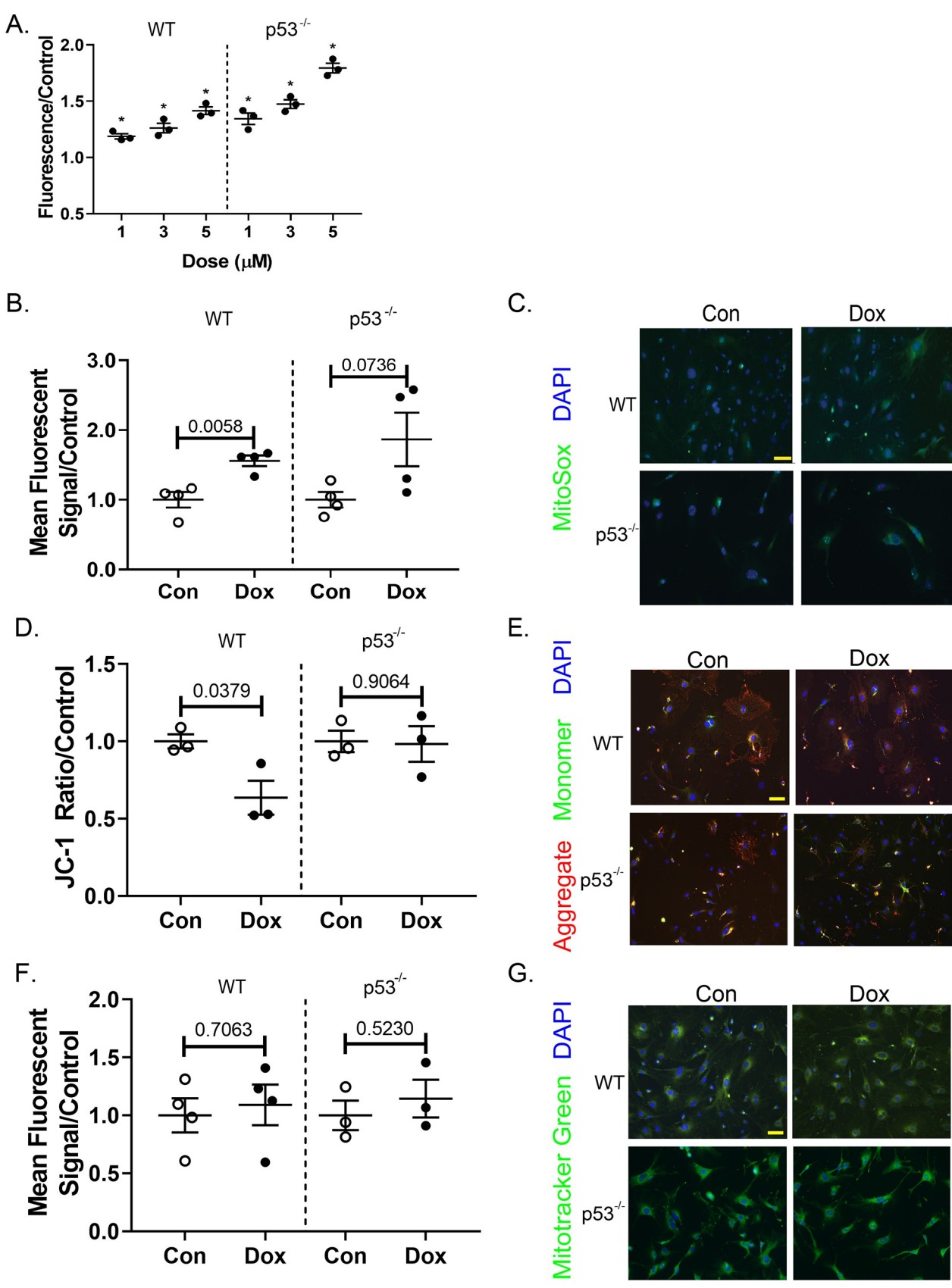

**Fig 2. DOX stimulates cardiac fibroblast mitochondrial ROS production and membrane depolarization in WT cells.** (A) DCFDA-stained cells were exposed to 1, 3, or 5 μL for 4 hrs. Fluorescence from ROS-reduced DCFDA was measured to determine relative ROS production. After standard treatment, cells were incubated with (B) MitoSOX Red, (D) JC-1, and (F) Mitotracker Green. Fluorescent signal for MitoSox Red and Mitotracker Green was measured via flow cytometry, while the two wavelengths for JC-1 were assessed using a spectrophotometer. (C), (E), and (G) are representative images of each stain, MitoSox, JC-1, and Mitotracker Green, respectively. Quantification includes at least 3 biological replicates and is represented as average ± SEM.

Mitochondrial function was assessed with the Agilent Seahorse Energy Phenotype Assay to determine if the increased oxidative stress and membrane depolarization caused mitochondrial dysfunction. Oxygen consumption (OCR) and extracellular acidification (ECAR) rates were obtained at baseline and after mitochondrial stressors. At baseline, a dose-dependent decrease in WT OCR was seen that was not seen in the ECAR baseline measurements. Despite the lower baseline OCR, there was not a statistically significant difference in percent OCR change (stressed OCR/baseline OCR) among the control and treated groups (**Fig 3A–3C**). There was a shift towards glycolysis in the treated cells in response to the mitochondrial stressors. The percent change was statistically significant in the 3 and 5 μM groups compared to control (**Fig 3E–3G**). p53$^{-/-}$ cells showed a similar dose-dependent decrease in OCR at baseline compared to control (**Fig 3B–3D**). After stress, the DOX-treated cells were able to increase mitochondrial respiration similar to control cells. At the 3 and 5 μM doses, samples actually showed a slight increase in metabolic potential compared to control cells (p = 0.005 and 0.0002, respectively). The increased metabolic potential indicates that DOX exposure primed the cells for increased mitochondrial respiration once under stress. The glycolytic shift seen in WT cells compared to control cells after stress was not seen in the p53$^{-/-}$ cells (**Fig 3F–3H**).

ROS production was increased in both cell strains at a cellular level. However, the same was not true at a mitochondrial level. ROS associated with the mitochondria was increased in WT cells compared to p53$^{-/-}$ cells. This could be due to early protection of mitochondria from DOX damage, or the expeditious clearance of dysfunctional mitochondria. ROS can induce mitochondrial dysfunction and depolarization, but dysfunctional mitochondria can also add to the presence of ROS. These changes are also reflected by the ability of mitochondria to produce energy via oxidative phosphorylation. We demonstrated, that after DOX exposure p53$^{-/-}$ utilized oxidative phosphorylation over glycolysis in response to mitochondrial stress. The opposite was true for WT cells. The following sections address one possible mechanism for the relationship between p53, mitochondrial dysfunction, and the altered CF phenotype.

## DOX induces mitophagy in cardiac fibroblasts

To test the cell's ability to undergo mitophagy in response to DOX, it was first demonstrated that the DOX-induced mitochondrial damage was sufficient to induce mitophagy. To assess mitophagy in primary CFs, we used a dual staining technique. **S1 Fig** demonstrates the two fluorescent dyes used to quantify lysosome content, mitophagy, and the overlap of these two components. DOX exposure increases the size and number of lysosomes compared to control (left column). There is only some overlap between the lysosomal dye and the mitophagy dye in the DOX samples. It is evident in the FCCP-treated sample that the two dyes co-localize with minimal lysosome-only dye visible. Increased lysosomes and Parkin expression (see **S1A Fig, Fig 5**) indicate that mitophagy is induced in DOX- and FCCP-treated cells. The area and intensity of the mitophagy dye in the p53$^{-/-}$ cells is increased in all treatment groups, including the control cells. The lysosomal dye is also more evident in all three treatment groups. This agrees with the flow cytometry data that indicated a high amount of fluorescent signal in all groups of the p53$^{-/-}$ for both the mitophagy dye and the lysosomal dye.

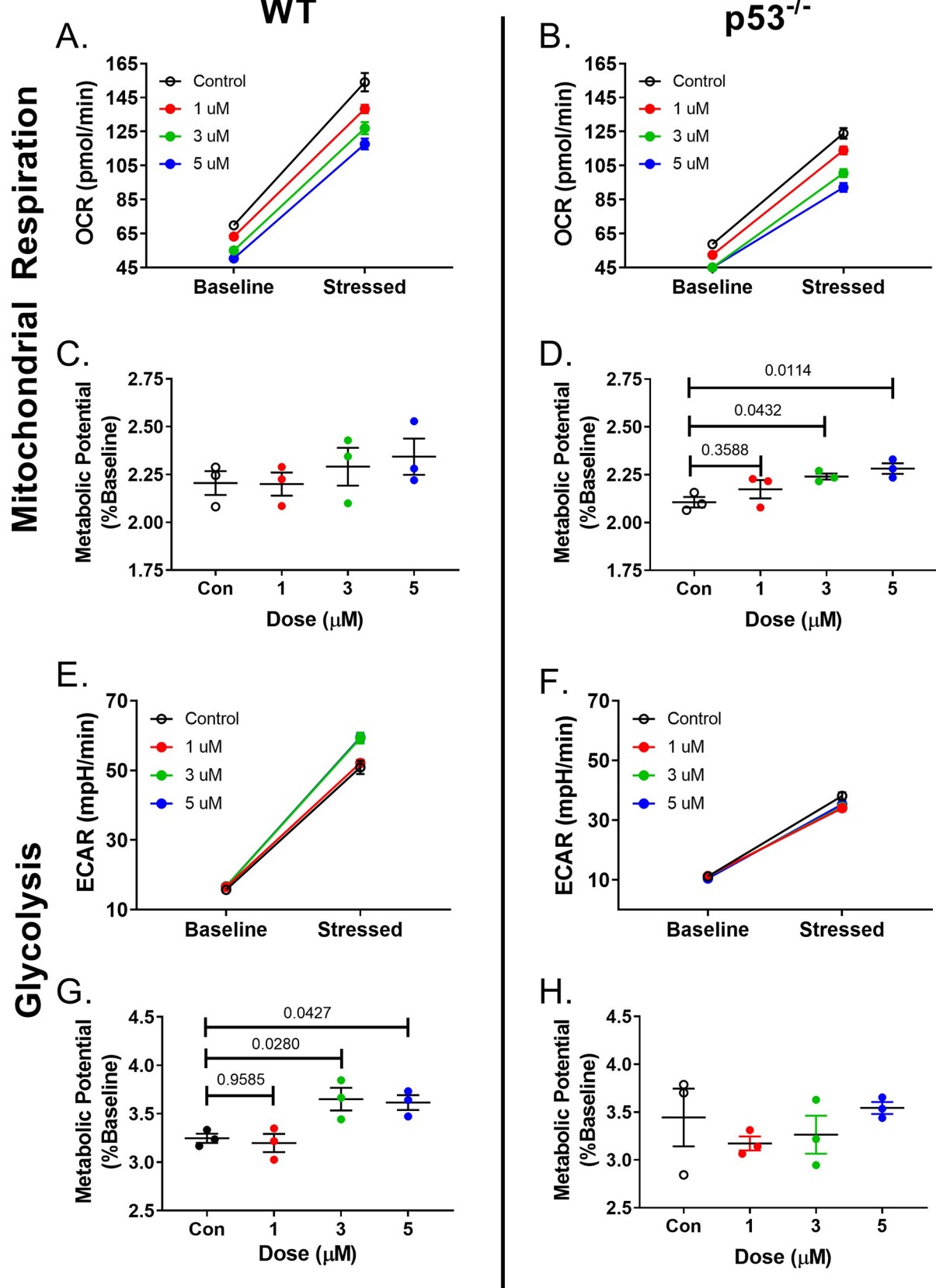

**Fig 3. DOX-induced mitochondrial dysfunction is partially ameliorated in p53$^{-/-}$ fibroblasts.** Oxygen consumption (A), (B), and ECAR via pH (E), (F) was measured before and after the addition of mitochondrial stressors in CFs previously exposed to DOX. (C), (D) demonstrate the metabolic potential of oxidative respiration in WT and p53$^{-/-}$ cells, respectively. Metabolic potential is a ratio of OCR after mitochondrial stressors over OCR at baseline. Metabolic potential of glycolysis was similarly calculated (G), (H).Quantification included 3 biological replicates and is represented as average ± SEM.

After confirming that mitophagy was initiated in response to DOX-induced mitochondrial damage, it was necessary to assess whether or not the cells were able to complete the process of mitophagy. This was measured by comparing the relative amount of overlap between the mitochondrial marker and lysosomal markers via flow cytometry. Before treatment with DOX or FCCP, cells were stained with a mitochondrial-binding dye. Under normal pH conditions, the mitochondrial dye exhibits fluorescence. However, in an acidic environment, as when an autophagosome and lysosome fuse, the fluorescent signal of the dye will increase substantially. The shift in fluorescent signal, along with co-localization of the dyes indicates the final step of mitophagy. To quantify this, separate samples were prepared for flow cytometry. The population of cells with high intensity signal for both wavelengths were considered "double positive" and indicative of an active mitophagy process. This quantification can be seen in **Fig 4D and 4E**. There was no significant difference between the control and DOX-treated WT samples in the percentage of double positive cells. However, there was an increase in double positive cells in the FCCP-treated fibroblasts compared to control and DOX samples (p = 0.03). Likely due to the overexpression of Parkin at baseline (**S2B Fig**), downstream processes including Parkin were upregulated. All treatment groups of p53$^{-/-}$ demonstrated a maximal increase in the intensity of the mitophagy marker. This demonstrates that in the absence of p53, these cells were able to carry out the process of mitophagy. PGC1α, an indicator of mitochondrial biogenesis, was similar between groups in both cell strains (**Fig 4F and 4G**). In the setting of mitochondrial dysfunction and mitophagy, PGC1α would be expected to increase. This may indicate another DOX-induced dysregulation in the mitochondrial quality control pathway.

## p53 prevents parkin localization

To understand the role of p53 in mitophagy, it was confirmed that p53 expression in primary CFs was increased after DOX exposure (**Fig 4A**). Not only did DOX increase p53 expression over two-fold compared to controls, it was not increased in FCCP-treated samples. This validates the use of FCCP as a positive control for mitophagy without the interference of DOX-induced p53 upregulation. Panels B and C of **Fig 4** demonstrate the gene mutation of p53$^{-/-}$ mice and the absence of p53 protein. The mutated gene has exons 2–6 deleted, which includes the translation starter site of the p53 gene. These mice begin developing tumors around four months of age, a time point after CF isolation.

**Fig 5** demonstrates the staining patterns of Parkin and p53 seen in control, DOX-, and FCCP-treated cells. In control cells, p53 remained diffuse throughout the cytoplasm. Parkin expression is not upregulated as it was with agents that induce mitochondrial dysfunction. Treatment with FCCP showed a distinct upregulation of Parkin and a small increase of p53. While p53 remained diffuse throughout the cell, the Parkin staining had a linear pattern, possibly due to localization to the mitochondria. In DOX-treated cells, p53 was substantially upregulated and much of it enters the nucleus. However, cytosolic p53 was present. Parkin was upregulated and remains diffuse throughout the cell with scant examples of linear patterning. There also appears to be nuclear localization of the Parkin in the DOX-treated samples. The increase in p53, or possibly some other DOX-induced change, may push Parkin towards its role as a transcription factor.

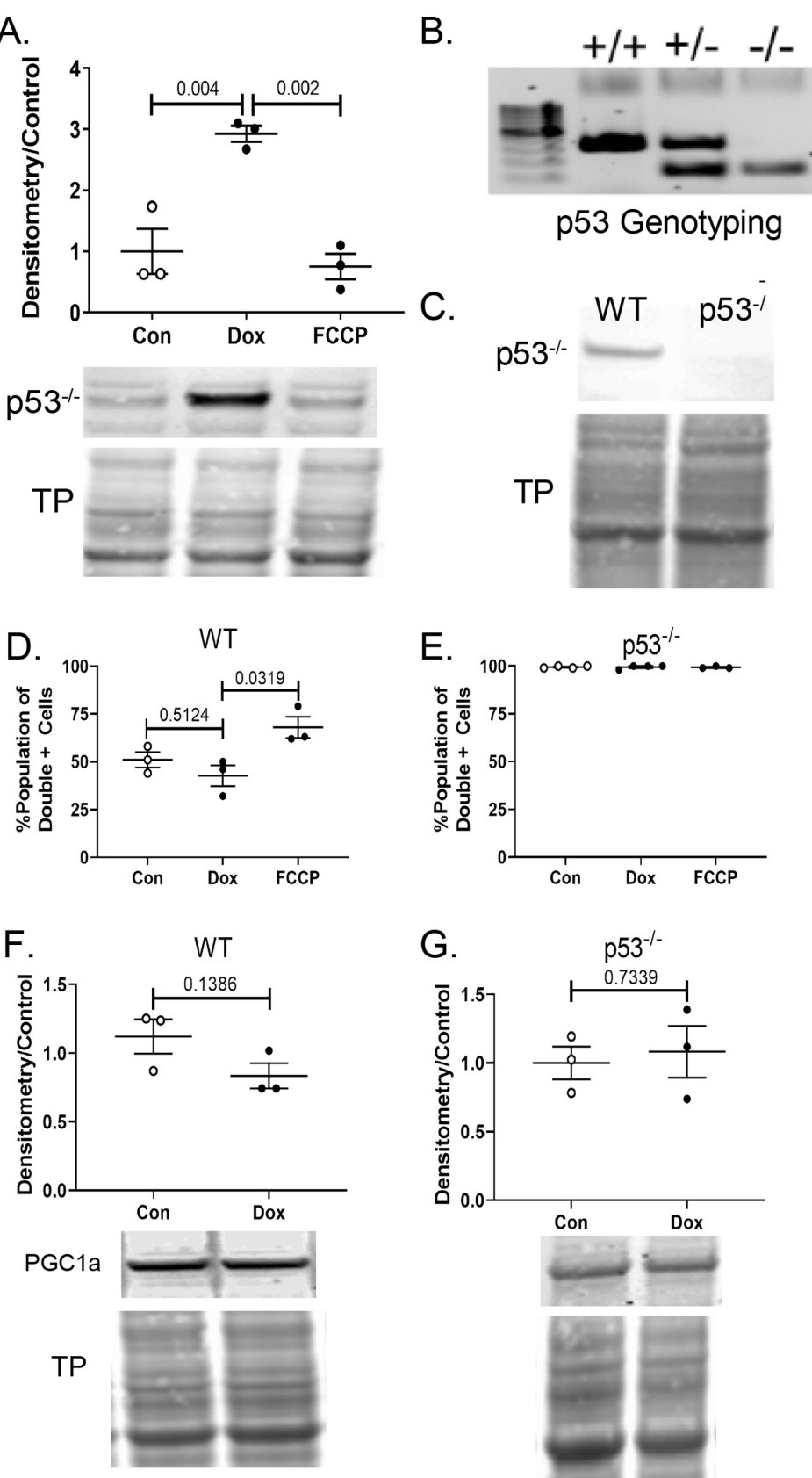

**Fig 4. Mitophagy and biogenesis balance.** (A) Western blot demonstrates relative protein content of p53 in WT control, Dox-exposed, and FCCP-treated cells with standard treatment. (B) Genotpying confirmed mice without the p53 allele and (C) null protein expression was verified via WB. (F), (G) PGC1α protein expression was measured using WBs. Densitometry of WB bands was obtained using image J and all samples were normalized to total protein content. Quantification includes 3+ biological replicates and is represented as average ± SEM.

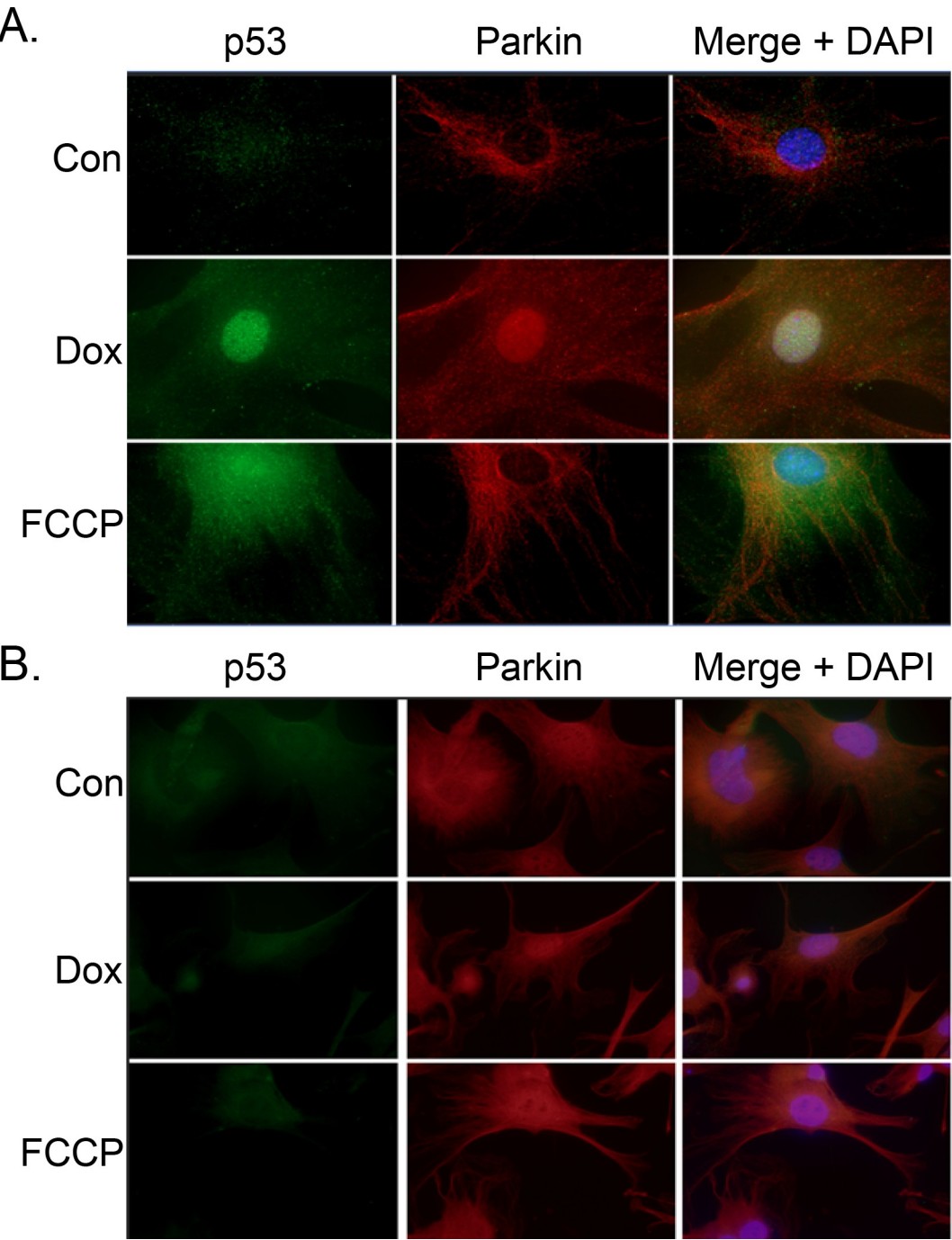

**Fig 5. Parkin and p53 exhibit different staining patterns in response to FCCP and DOX.** After standard treatment, WT (A) and p53$^{-/-}$ were stained with fluorescent p53 and Parkin antibodies. Parkin expression in p53$^{-/-}$ cells was upregulated and exposure time had to be decreased compared to WT cells to avoid oversaturation. 40X Magnification.

The most notable finding in the p53$^{-/-}$ cells was the overexpression of Parkin (**Fig 5B**). Exposure time had to be reduced to avoid oversaturation (5 s in WT samples, compared to 500 ms in p53$^{-/-}$). Increased Parkin was seen in cells exposed to DOC and FCCP, as well as control cells. The fluorescence signal in the green channel is the background fluorescence that is also visible in the WT cells, but obscured by the more intense puncta of the p53 staining. The Parkin staining pattern appears to show diffuse and linear patterning in each treatment group. This may indicate the localization of the Parkin to mitochondria, along with excess Parkin remaining at large throughout the cytoplasm. There also is some nuclear staining seen in each treatment group. This may be Parkin that has localized around the nucleus, or into the nucleus. Parkin can also act as a transcription factor and the overexpression may lead to increased nuclear localization, even at baseline.

To determine if the p53:Parkin interaction was the cause of the reduced mitophagy in DOX-treated cells, we assessed if Parkin was able to translocate to the mitochondria after DOX treatment and the interaction between p53 and Parkin. After treatment with DOX or FCCP, cell samples were processed to yield a cytoplasmic fraction and a mitochondrial fraction. Immunoblotting with anti-Parkin showed that Parkin was present in the cytoplasm of the control, DOX, and FCCP samples. Minimal Parkin was present in the mitochondrial fraction of the control and DOX samples (**S2C Fig**). The Parkin present in the mitochondrial fraction of the FCCP sample had a molecular weight shift up, indicating a slight increase in molecular weight. For Parkin to locate to the mitochondria, it is first ubiquitinated and then activated via phosphorylation when it reaches the mitochondrial membrane. This finding agrees with similar studies that demonstrate an increase in the molecular weight of mitochondrial-associated Parkin due to monoubiquitination [36]. Parkin levels in the whole cell and cytoplasm of p53$^{-/-}$ cells appear equal due to a higher basal level of Parkin (**S2B–S2D Fig**). Parkin with the molecular weight shift was observed in the mitochondrial fraction of the p53$^{-/-}$ cells treated with DOX.

To determine if p53 was binding to Parkin, we used a proximity ligation assay. Fixed primary CFs were probed with anti-p53 and anti-Parkin antibodies. PLUS and MINUS probes capable of forming circular DNA constructs were pre-conjugated to the secondary antibodies. Samples were incubated with ligase to create the circular DNA. Lastly, the circular oligonucleotides are amplified with fluorescent labels. Increased fluorescent puncta indicates an increase in the p53:Parkin interaction. Proteins must be within 40 nm of each other for ligation of the circular DNA to occur. DOX-treated cells showed a greater number of fluorescent puncta and increased fluorescent intensity compared to control cells (**Fig 6B and 6C**). The granular staining pattern observed in the images is indicative of the technique used to detect protein:protein interactions. The perinuclear staining pattern may be due to the roles of p53 and Parkin [37] (alves de Costa 2019) as transcription factors. Additionally, mitochondria will exhibit perinuclear localization under specific circumstances [38, 39]. The localization of the proteins explains the perinuclear staining. Fluorescent intensity was quantified using Image J (**Fig 6A**) and demonstrated an increase in fluorescent signal in cells treated with DOX. Negative controls showed a small amount of non-specific signal in the DOX samples processed with anti-Parkin antibody, but not in anti-p53 primary only controls. In p53$^{-/-}$ fibroblasts there was an increase in the background of these same negative controls.

Dox exposure leads to the development of ROS and mitochondrial dysfunction. Cells attempt to clear the dysfunctional mitochondria by initiating mitophagy. However, in WT cells, DOX also causes cell stress that leads to the upregulation of p53. Parkin is prevented from reaching the mitochondria and signaling for lysosomal engulfment by p53 sequestration. Many of these changes are not seen in the p53$^{-/-}$ cells as Parkin is now free to continue the process of mitophagy, thereby relieving the mitochondrial stress of the cell.

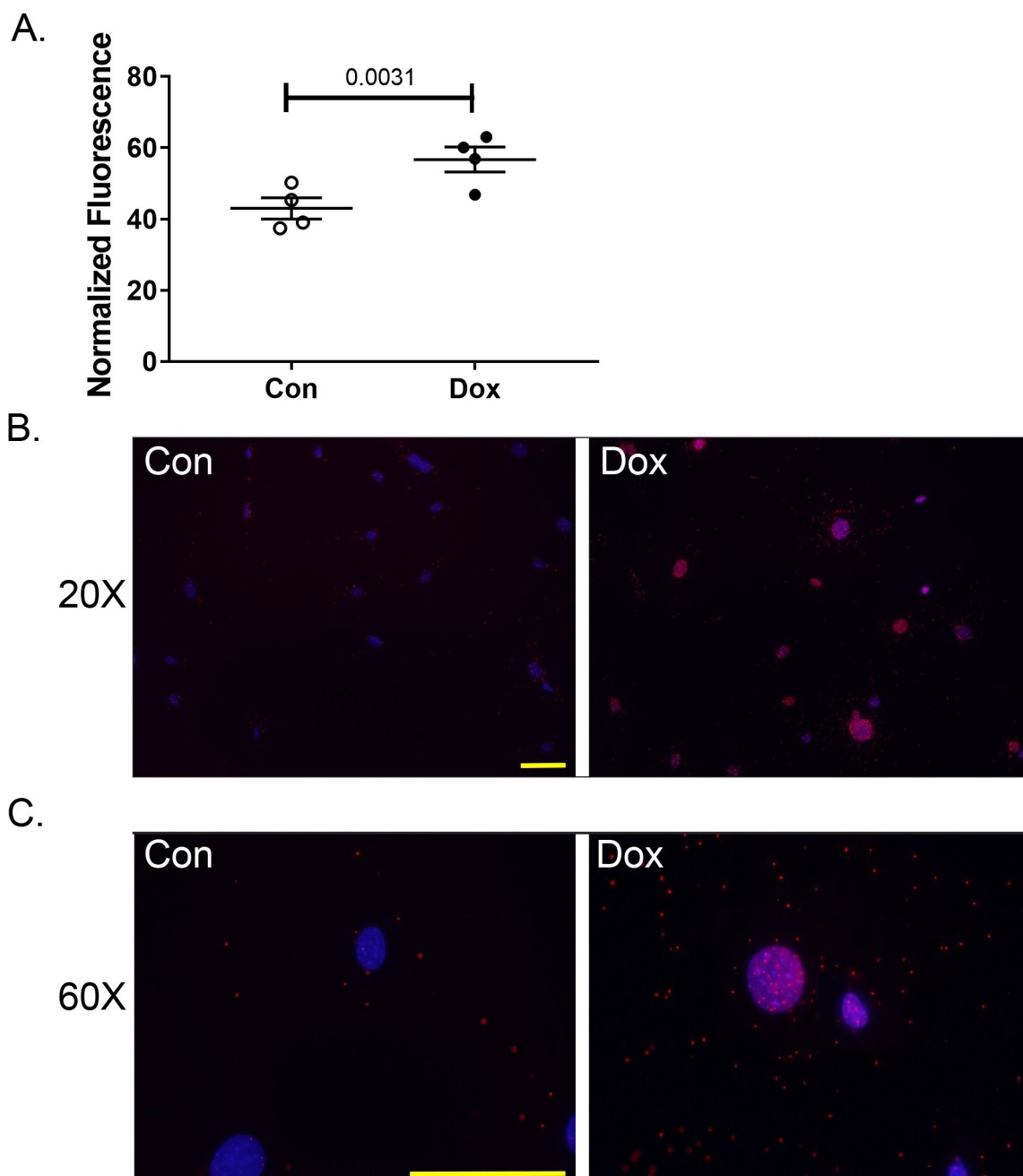

**Fig 6. DOX increases Parkin: p53 interactions.** (B, C) A proximity ligation assay demonstrates the increased interactions between the Parkin and p53 proteins in WT cells exposed to DOX. (A) Quantification of fluorescent signal. Quantification consists of 3–4 biological replicates with at least 10 fields of view acquired at 20X magnification. Samples were normalized to nuclear stain. Average ± SEM.

## Discussion

Healthy CFs are essential for normal cardiac performance and the ability to respond to stressors and injury. While integral to directing injury response to myocardial infarctions and other forms of cardiac damage, CFs are also essential for proper electrical conduction in the heart, as they regulate calcium homeostasis through myocyte-fibroblast coupling [40, 41]. Additionally, CFs maintain tissue structure by regulating the deposition and remodeling of collagen and

other ECM components [42, 43]. Considering the various functions of CFs in normal and pathologic states, understanding their role in DOX cardiotoxicity is crucial to developing preventive or therapeutic treatments. We have identified one mechanism of DOX damage that focuses on the CF, demonstrating the potential significance of the CF in the progression of DOX cardiotoxicity. These observations are summarized in **Fig 7**.

DOX targets tumor cells via DNA intercalation and inhibition of topoisomerase II, disrupting DNA and macromolecular synthesis, and triggering double strand DNA breaks [44, 45]. Prior research has established the pathological processes triggered by DOX in cardiac myocytes. The prevailing theory for myocyte damage is centered on DOX induction of ROS [7], which leads to mitochondrial damage and apoptosis [46]. Due to high mitochondrial load and a lack of scavenger molecules, cardiac myocytes are more prone to oxidative damage than other cell types [47]. Cardiac myocytes are dependent on oxidative phosphorylation to meet the energy demands of the cell, while tumor cells utilize glycolysis, even in an aerobic environment. The differences in energy sources and proliferation dictate the differential effects of DOX on these cell types but do not fully explain the effects of DOX on CFs.

In response to DOX, cells throughout the body upregulate expression of the tumor suppressor p53 [48–50]. In the heart, this occurs in cardiac myocytes, and as our data shows, in CFs. In response to DNA damage, p53 undergoes a translocation to the nucleus to initiate a combination of cell cycle arrest, DNA damage repair, or apoptosis. In our model, p53 causes cell cycle arrest that does not progress to apoptosis in the timespan studied. The DOX-induced phenotype observed in this study did not indicate an activation to a myofibroblast. There were similarities to a senescent phenotype, such as arrested proliferation and increased cytokine production. It is likely that a subgroup of the cell population does enter senescence, however the morphology of the cells does not support this phenotype for the majority of the culture, in either WT or p53$^{-/-}$. WT cells demonstrated a DOX-induced decrease in proliferation, likely secondary to p53-dependent cell cycle arrest. A similar decrease in proliferation was seen in the p53$^{-/-}$ cells but without the concomitant cell cycle arrest. There are a number of factors that could play into the reduced proliferation of the p53$^{-/-}$ cells. DOX can alter gene expression through multiple mechanisms, such as histone eviction and redistribution [51, 52]. One study demonstrated "metabolic re-wiring" with suppression of pathways necessary for the G1 phase of the cell cycle [53]. More pertinently, inhibition of mitophagy has been used as way to induce cellular apoptosis [54, 55]. Mitochondrial dysfunction in a cell without the ability to undergo mitophagy, as seen in the WT cells, will trigger cell stress pathways leading to cell cycle arrest and eventually apoptosis. However, by inducing mitophagy groups have shown decreases in cell proliferation unrelated to an arrest in cell cycle [56, 57]. Multiple mechanisms have been proposed, including the simple explanation that cellular energy must be rerouted to the processes of mitophagy and mitochondrial biogenesis.

*Zhan et al*. demonstrated an increase in ataxia telangiectasia mutated (ATM) protein after DOX exposure, indicating a DNA damage response. *Zhan et al*. chose to study CFs because the increase in ATM was predominantly in CFs, as opposed to other cells of the heart, such as cardiac myocytes [58]. This data further supports the idea that DNA damage and mitochondrial damage are both significant stresses to the CF.

From immunocytochemistry it is evident that an appreciable amount of p53 remains in the cytosol. It is a commonly seen phenomenon that when a cell is under stress, mitochondria take on a peri-nuclear localization. In Fig 6, much of the interaction between the p53 and Parkin seems to be within or around the nucleus. As Parkin can act as a transcription factor and enter the nucleus, this interaction could be happening in the nucleus. It is also possible that as the mitochondria take on a peri-nuclear pattern, the Parkin attempting to bind to the mitochondrial membrane also take on a peri-nuclear pattern. Whether within the nucleus or without,

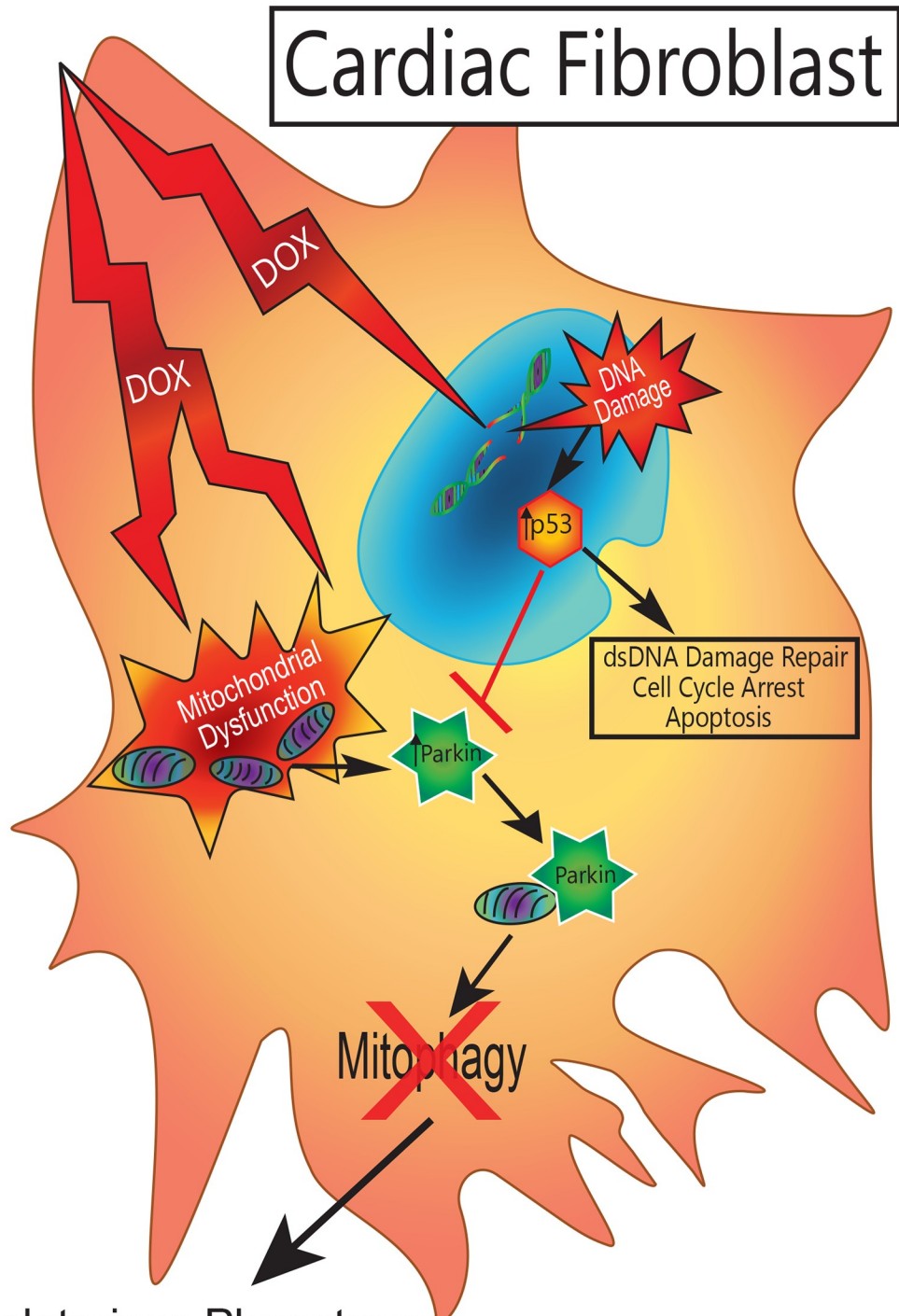

**Fig 7. Parkin/p53 interference.** DOX induced a deleterious phenotype in WT CFs. Due to the cell's inability to respond to multiple stressors, an early DCM gene profile was adopted. This included increased cardiac remodeling genes and increased inflammatory signaling. The phenotype also removes the cell's ability to respond to injury by inhibiting proliferation and migration. Lastly, the cell is open to increased damage as it is unable to respond to metabolic stressors due to mitochondrial dysfunction and deficits in clearing the dysfunctional mitochondria.

p53 interaction with Parkin is increased after DOX exposure. *Hoshino et al.* described the direct interaction between cytosolic p53 and Parkin in cardiac myocytes that prevented Parkin from binding at the mitochondrial membrane [59]. They also demonstrated that this interaction was sufficient to disrupt the process of mitophagy in cardiac myocytes. In fact, whole-body p53 depletion in mouse models attenuated cardiac dysfunction and decreased fibrosis after DOX exposure compared to wild type mice [49]. Curiously, when p53 deletion was restricted to cardiac myocytes, the same benefits were not seen [60]. These factors suggest that DOX may elicit a class of pathological effects in CFs that does not resemble what is seen in either myocytes or tumor cells.

Mitophagy at homeostasis is generally thought to be mediated by BNIP3. However, studies have shown that both tumor cells and cardiac myocytes can utilize the BNIP3 pathway under stress [61, 62]. In the absence of Parkin, these cell types may be able to maintain mitophagy under stress. The role of BNIP3 in relation to CFs is not well studied. One lab demonstrated that mitophagy in CFs can be carried out with BNIP3, but under conditions of BNIP3 overexpression and still involving Parkin [63]. Increased mitophagy is balanced by increased mitochondrial genesis, a process mediated by PGC1α expression. CFs did not exhibit a concomitant increase in PGC1α, indicating another process of mitochondrial homeostasis that may be affected by DOX exposure.

WT cells and p53$^{-/-}$ cells demonstrated dose-dependent increases of ROS after exposure to DOX. WT cells exposed to DOX also displayed a uniformly higher intensity of fluorescent signal from mitochondrial ROS. In p53$^{-/-}$ cells exposed to DOX, the signal was not as uniform, with about half of the samples showing no changed from control cells. Even though DOX exposure ramped up ROS production in both cell types, the p53$^{-/-}$ cells fared better. We believe that this was at least in part due to mechanism we have proposed. However, DOX, with or without ROS, can cause damage to numerous, if not all, organelles in a cell [64–69]. Our study focused on the mitochondria with a potential mechanism that had been observed under specific circumstances in other studies. Future research could investigate the significant changes in p53$^{-/-}$ cells that may be beneficial to other organelles.

Based on the gene expression changes in DOX-exposed CFs quantified in this study, it is possible that a significant subset of cells are undergoing senescence. However, a recent study described a "survival" phenotype that CFs undergo in response to oxidative stress [70]. Regardless of how the cells are characterized, DOX-exposed CFs will likely be unable to respond adequately to future cellular and cardiac stresses. With a decreased ability to proliferate and migrate, as noted in our data, the heart will be dependent on non-cardiac sources of fibroblasts if injury arises. Furthermore, in a heart that is not fully mature, i.e. a pediatric patient, the DOX-induced CF phenotype may alter the maturation of the heart. Our data suggest that early changes in the gene expression promote ECM remodeling, with an increase in remodeling genes and a decrease in structural genes. While CFs are directing cardiac remodeling during maturation, they also participate in signaling myocyte hypertrophy. The effects of increased inflammatory cytokines in the microenvironment on long-term cardiac function are not fully known but are associated with dilated cardiomyopathy (DCM). Disruptions to this process could have far-reaching implications on cardiac function.

Of the 32 genes upregulated by DOX in WT CFs, there were a few of note. Interestingly, C-X-C Motif Chemokine Ligand 10 (Cxcl10), and CxCl11 proteins were both upregulated in patients with heart disease [71, 72] and some studies indicate this is true for DOX cardiotoxicity as well [73, 74]. Multiple studies showed that cadherin-2 (Cdh2) increased in dilated cardiomyopathies [75, 76]. Its role as a possible biomarker for DOX cardiotoxicity has not been investigated, but one study showed increased gene expression in DOX-exposed MCF-7 breast cancer cells [77]. An early response to DOX-induced damage in the CFs of patients may be the release of Cxcl10, Cxcl11, or Cdh2. These molecules should be further investigated as clinical biomarkers of DOX-induced cardiac disease.

Historically, CFs have not been the focus of research on the detrimental effects of DOX on the heart. Therefore, DOX's effect on CFs offers fertile ground for discovering novel ways to better protect the myocardium during and after DOX treatment. The use of small-molecule p53 inhibitors are currently under investigation in clinical trials for cancer therapy. In tumors without p53 deletions or mutations, p53 can be protective [78]. By inducing cell cycle arrest, p53 will induce apoptosis or prevent the tumor cell from proliferating. In other cases, gain-of-function mutant p53 forms can interfere with autophagy and promote anti-apoptotic pathways [79]. In these instances, the mutant p53 actually promotes tumor proliferation. Theoretically, by selectively knocking down p53 expression during cancer treatment in specific tumors, you may sensitize the tumor to chemotherapy while protecting the CFs, and thereby the heart, from disrupted mitophagy. Additionally, just as there is research into targeted delivery of chemotherapeutic agent to tumors, research could look into the targeted delivery of protective agents to the heart. In this way, patients who would not benefit from a p53-inhibitory treatment could still benefit from the possible cardioprotective effects of p53 inhibition. Other therapeutic options include salvaging mitophagy by replacing or bypassing Parkin. The BNIP3 pathway, thought to be more active during homeostatic mitophagy, could be stimulated to offset the Parkin sequestration.

The current study is limited by the use of mono-cultured cells. CFs and cardiac myocytes each highly influence the behavior of the other. Future studies will include adapting the research to co-cultures of CFs and myocytes to enhance the understanding of cell-cell communication and determine the effect of the dysfunctional fibroblast on myocyte health. An in-depth look at the role of mitochondrial dysfunction in the CF is also necessary. Investigating retrograde nuclear signaling among other mitochondrial signaling mechanisms will help determine the exact connection between mitochondrial dysfunction and the DOX-induced phenotype. A purposeful limitation to this study was the use of isolated CFs. We isolated CFs from our WT and p53$^{-/-}$ before exposing them to DOX to study the role of p53 solely in CFs. Our p53$^{-/-}$ mouse strain is a whole body knock out. We wanted to isolate the role of p53 in CFs, as opposed to changes caused by other p53 deficient cells. We chose to perform these initial studies in vitro to support our hypothesis. Our lab is currently developing mice that have an inducible KO of p53 that will only be present in cardiac fibroblasts. These studies have demonstrated that DOX does alter CF function and this has broad implications on the cardiac microenvironment and therefore cardiac function. To date, no study has investigated the effect of DOX exposure on CFs in depth. The novel findings indicate an important role for fibroblasts to play in DOX cardiotoxicity that may be mediated by mitochondrial health and quality control. Further study of the CF will increase our understanding of the disease progression and create new avenues for therapeutic targets. If phenotypic changes in the CF are an early indicator of cardiac toxicity, it is also possible that associated biomarkers may be developed to screen cancer drugs in development for cardiotoxic effects. If further study of the cardiac fibroblast can help protect the heart during DOX exposure, we will be able to improve the quality of life for the ever-growing population of childhood cancer survivors.

## Supporting information

**S1 Fig. Representative images of lysosome/mitophagy staining in WT cells.** WT (A) and p53$^{-/-}$ (B) cells were stained with a mitochondrial dye before treatment and a lysosomal dye after treatment. Fluorescent intensity of the mitochondrial dye increases when it is exposed to an acidic pH. Magnification 60X.
(TIF)

**S2 Fig. Parkin cannot localize to damaged mitochondria after DOX exposure.** (A) Fraction specificity confirmed with subcellular specific antibodies. (B) Parkin expression, via WB, in WT and p53$^{-/-}$ after standard treatment. In (C) and (D), cells underwent standard treatment and at protein extraction separated into subcellular fractions according to kit protocol. The fractions were then separated using gel electrophoresis and probed for the presence of Parkin. Parkin was observed in the mitochondrial fraction after DOX exposure. Quantification include 3 biological replicates and is represented as average ± SEM.
(TIF)

**S1 Raw images.**
(PDF)

## Author Contributions

**Conceptualization:** T. R. Mancilla, G. J. Aune.

**Data curation:** T. R. Mancilla, L. R. Davis, G. J. Aune.

**Formal analysis:** T. R. Mancilla, G. J. Aune.

**Funding acquisition:** T. R. Mancilla, G. J. Aune.

**Investigation:** T. R. Mancilla, G. J. Aune.

**Methodology:** T. R. Mancilla, L. R. Davis, G. J. Aune.

**Project administration:** G. J. Aune.

**Resources:** T. R. Mancilla.

**Software:** T. R. Mancilla, G. J. Aune.

**Supervision:** G. J. Aune.

**Validation:** T. R. Mancilla, L. R. Davis, G. J. Aune.

**Visualization:** T. R. Mancilla, L. R. Davis, G. J. Aune.

**Writing – original draft:** T. R. Mancilla, G. J. Aune.

**Writing – review & editing:** L. R. Davis, G. J. Aune.

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
