## [Decision Letter · Decision Letter 0]

8 Apr 2020

PONE-D-20-07599

Doxorubicin-induced p53 Interferes with Mitophagy in Cardiac Fibroblasts

PLOS ONE

Dear Dr. Aune,

Thank you for submitting your manuscript to PLOS ONE. After careful consideration, we feel that it has merit but does not fully meet PLOS ONE’s publication criteria as it currently stands. Therefore, we invite you to submit a revised version of the manuscript that addresses the points raised during the review process.

Your manuscript was reviewed by two experts and both of them provided detailed comments.Please address all comments as appropriate.

We would appreciate receiving your revised manuscript by May 23 2020 11:59PM. To enhance the reproducibility of your results, we recommend that if applicable you deposit your laboratory protocols in protocols.io, where a protocol can be assigned its own identifier (DOI) such that it can be cited independently in the future. For instructions see: http://journals.plos.org/plosone/s/submission-guidelines#loc-laboratory-protocols

We look forward to receiving your revised manuscript.

Kind regards,

Partha Mukhopadhyay, Ph.D.

Academic Editor

PLOS ONE

Journal Requirements:

2) To comply with PLOS ONE submissions requirements, in your Methods section, please provide the number of animals used in this study.

3) PLOS ONE now requires that authors provide the original uncropped and unadjusted images underlying all blot or gel results reported in a submission’s figures or Supporting Information files. This policy and the journal’s other requirements for blot/gel reporting and figure preparation are described in detail at https://journals.plos.org/plosone/s/figures#loc-blot-and-gel-reporting-requirements and https://journals.plos.org/plosone/s/figures#loc-preparing-figures-from-image-files. When you submit your revised manuscript, please ensure that your figures adhere fully to these guidelines and provide the original underlying images for all blot or gel data reported in your submission. See the following link for instructions on providing the original image data: https://journals.plos.org/plosone/s/figures#loc-original-images-for-blots-and-gels.

Reviewers' comments:

Reviewer's Responses to Questions

**Comments to the Author**

1. Is the manuscript technically sound, and do the data support the conclusions?

Reviewer #1: Partly

Reviewer #2: Yes

2. Has the statistical analysis been performed appropriately and rigorously? 

Reviewer #1: Yes

Reviewer #2: Yes

3. Have the authors made all data underlying the findings in their manuscript fully available?

Reviewer #1: Yes

Reviewer #2: Yes

4. Is the manuscript presented in an intelligible fashion and written in standard English?

Reviewer #1: Yes

Reviewer #2: Yes

5. Review Comments to the Author

Reviewer #1: The current study "Doxorubicin-induced p53 Interferes with Mitophagy in Cardiac Fibroblasts" discussed the effect of p53 deficiency on Doxorubicin induced mitochondrial change. Here are some major concerns:

1. The authors need to better arrange the data presentation and result description. Many closely related figures should be combined together. Conclusions in each part should be clearly given. Also, the authors should be more careful about the figure number and panel consistency.

2. The current study points out that DOX compromises mitochondria in the wild type cells but not the p53 deficient cells. More than half of the data are the consequences of this phenotype, including ROS, metabolism change and mitophagy increase in the wild type cells. Have the authors researched into the reason p53 deficient cells are resistant to the DOX mediated mitochondria damage?

3. The authors found that DOX treated p53 deficient cells have similar proliferation phase pattern compared to the non-treated mice, which is an interesting phenotype compared to WT cells. However, DOX really inhibits the cell proliferation in both cells. Have the authors discussed the mechanism?

4. Western blot in the same assay with all samples is needed to confirm the upregulation claim of Parkin in p53 deficient cells.

5. “Difficult to assess” and “not three repeats” should not be used to cover the unshown or unexplained data

6. If Parkin is not upregulated in the WT cell mitochondria in DOX treatment, how do the authors explain the upregulated mitophagy?

7. In figure 12, the interaction of Parkin and p53 is mostly in the nucleus. How is it related to mitophagy? Also, FCCP treatment should be put in Figure 12 too.

8. Judging from the current data, p53 deficiency protects the CF cells from the very beginning of DOX treatment. The authors need to make the conclusions more carefully.

Reviewer #2: The manuscript entitled “Doxorubicin-induced p53 interferes with mitophagy in cardiac fibroblasts” suggested the effect of doxorubicin on cardiac fibroblast functions and the involvement of p53. In detail, doxorubicin-induced damage on cardiac fibroblasts was correlated with decreased proliferation/migration, and cell cycle arrest, whereas those changes were recovered back in P53 deficient fibroblast. Based on accumulated results, the authors clearly demonstrated that cardiac fibroblasts are an important myocardial cell type to investigate in the context of doxorubicin treatment. This article is very straightforward to understand the novel role of p53 in myocardial cells. However, there are several concerns must be addressed.

1) The authors used isolated fibroblasts from WT and P53 deficient mouse for in vitro experiments. It should be better to check the Dox-induced cardiac toxicity and involvement of P53 pathway in vivo. If the in-vivo datas are included with evaluation of DNA damage, mitochondrial dysfunction, cell cycle arrest, cell death on cardiac cells after Dox IP injection on WT and P53 KO mice, that will strengthen the authors’ ideas.

2) The authors described the results obtained in figure legends. In figure legend is only for demonstrating the methodology how to conduct experiment. Please re-phase figure legends.

3) Table1 and 2, it will be better to re-organize these tables to hitmap. And try to merge Table 1 and Table 2 for easy understanding and after merging re-organize merged information to hitmap.

4) In Figure2B & C, it is recommended to show actual FACS plot data (including the percentage of G1,S and G2) for better understanding at a glance.

5) There is too much information in Materials & Methods. It will be better to make compact.

6) There are some minor errors/mistakes in manuscript and figures. Please re-check them very carefully and thoroughly. Some identified was shown below.

- In Figure 1, panel C and D are missing.

- In Figure3, put “WT” on the top of Panel A,C,E column and put “P53-/-” on the top of Panel B,D,F column.

- In Figure4C, the cell number between groups is too different. Show the representative images which have similar cell number.

- In Figure5A and B, Put “mitotracker” and “JC-1” on the top of panel A and B for better understanding.

- In Figure5C, put the full name of MTG to mitotracker green for easy understanding.

- In Figure6 and manuscript, the authors used terms “baseline” and “stressed”. It might be better to change “con” and “Dox”.

- In Figure9B and C, it will be better to move to supporting data.

- In Figure 10A, it will be better to move to supporting data.

- In Figure 10C, graph (mitochondrial) is missing.

- In Figure11, indications for A and B are missing.

- Page 9, paraformaldehyde, put the % of paraformaldehyde (might be 4%).

- Page11, what is the pore size of Boyden chamber?

- Page 15 line11, what is the dose of DOX?

6. PLOS authors have the option to publish the peer review history of their article (what does this mean?). If published, this will include your full peer review and any attached files.

Reviewer #1: No

Reviewer #2: No

---

## [Author Response · Author response to Decision Letter 0]

1 Jun 2020

Dear Reviewers:

Thank you for the thoughtful and clear critiques of our manuscript. Below we have responded to each of the points raised. For clarity, each review point is included in black text and our response in blue text.

Reviewer #1: 

The current study "Doxorubicin-induced p53 Interferes with Mitophagy in Cardiac Fibroblasts" discussed the effect of p53 deficiency on Doxorubicin induced mitochondrial change. Here are some major concerns:

1. The authors need to better arrange the data presentation and result description. Many closely related figures should be combined together. Conclusions in each part should be clearly given. Also, the authors should be more careful about the figure number and panel consistency. Figures have been combined to reduce the overall number of figures and careful attention was paid to the panel labeling in the construction of the figures. Conclusion statements were added to the results sections. Through this process we have taken the 13 figures in the original submission and condensed them to 7 primary figures with two supplementary figures.

2. The current study points out that DOX compromises mitochondria in the wild type cells but not the p53 deficient cells. More than half of the data are the consequences of this phenotype, including ROS, metabolism change and mitophagy increase in the wild type cells. Have the authors researched into the reason p53 deficient cells are resistant to the DOX mediated mitochondria damage? While there are most likely multiple mechanisms related to the p53-/- attenuation of DOX toxicity in CFs, we explored a specific mechanism wherein p53 upregulation from DNA damage and cellular stress interferes with physiological and pathological mitochondrial quality control. Mitochondrial quality control includes the process of Parkin-dependent mitophagy. We found that p53 sequesters Parkin in the CFs preventing the clearance of dysfunctional mitochondria through the mitophagy process. In the absence of this sequestration by p53, Parkin is free to mediate removal of dysfunctional mitochondria and results in the phenotype noted in p53-deficient cells.

3. The authors found that DOX treated p53 deficient cells have similar proliferation phase pattern compared to the non-treated mice, which is an interesting phenotype compared to WT cells. However, DOX really inhibits the cell proliferation in both cells. Have the authors discussed the mechanism? The possible mechanism is only briefly discussed in this manuscript as it does not appear to be related to p53 cell cycle regulation in the p53-/- cells. For this manuscript we wanted to focus more on the changes in mitophagy.

4. Western blot in the same assay with all samples is needed to confirm the upregulation claim of Parkin in p53 deficient cells. This is now included in supplemental figure S2B

5. “Difficult to assess” and “not three repeats” should not be used to cover the unshown or unexplained data This data was moved to the supplemental section and that language removed.

6. If Parkin is not upregulated in the WT cell mitochondria in DOX treatment, how do the authors explain the upregulated mitophagy? Mitophagy is not upregulated in the WT cells after exposure to DOX. The level of mitophagy is similar to that of the WT control cells.

7. In figure 12, the interaction of Parkin and p53 is mostly in the nucleus. How is it related to mitophagy? Also, FCCP treatment should be put in Figure 12 too. Both p53 and Parkin are present in the cytoplasm but can cross the nuclear membrane to act as transcription factors. Additionally, under cell stress, mitochondria often take on a perinuclear localization pattern. The p53/Parkin interaction may be taking place inside the nucleus or just outside the nuclear membrane. Importantly, our data shows that this interaction does occur and is increased after DOX exposure. We did not include FCCP in Figure 12 from the original manuscript because our experiments indicated that it does not robustly induce p53 expression.

8. Judging from the current data, p53 deficiency protects the CF cells from the very beginning of DOX treatment. The authors need to make the conclusions more carefully. We took this comment into account and throughout our revisions made a point to draw our conclusions based upon the data we present and to not go beyond these included experiments.

Reviewer #2

1. The authors used isolated fibroblasts from WT and P53 deficient mouse for in vitro experiments. It should be better to check the Dox-induced cardiac toxicity and involvement of p53 pathway in vivo. If the in-vivo datas are included with evaluation of DNA damage, mitochondrial dysfunction, cell cycle arrest, cell death on cardiac cells after Dox IP injection on WT and P53 KO mice, that will strengthen the authors’ ideas. p53 KO mice cannot live beyond 3 months without developing tumors, due to full body knockout of this critical tumor suppressor gene. We isolated cardiac fibroblasts before exposing them to DOX to study the role of p53 solely in CFs. We chose to perform these initial studies in vitro to test our hypothesis. Our lab is currently developing mice that have an inducible KO of p53 that will only be present in cardiac fibroblasts. Using these mice important questions such as the point raised here can be addressed. However, for this manuscript we felt these experiments were beyond the scope of our initial efforts to establish DOX-induced mitochondrial damage in cardiac fibroblasts as a damage mechanism with the potential to cause cardiac pathology.

2. The authors described the results obtained in figure legends. In figure legend is only for demonstrating the methodology how to conduct experiment. Please re-phase figure legends. Figure legends were modified to only reflect methods.

3. Table1 and 2, it will be better to re-organize these tables to hitmap. And try to merge Table 1 and Table 2 for easy understanding and after merging re-organize merged information to hitmap. Attempts were made to combine and reconfigure the data into a heat map that was intuitive to read. However, we were not satisfied that the end products were useful. Since many of the genes that were differentially regulated differed between the WT and KO mice, we did not combine the tables in their current formats. In our opinion, we believe that presenting this data in concise table format is superior because we can present the exact fold change in a way that is easy to view quickly.

4. In Figure2B & C, it is recommended to show actual FACS plot data (including the percentage of G1,S and G2) for better understanding at a glance. This figure was kept “as is” because we prefer to show the average and SEM for the biological replicates. This is not an uncommon way to show this data as this type of depiction is frequently found in the scientific literature.

5. There is too much information in Materials & Methods. It will be better to make compact. Revisions for clarity and brevity were made where possible. A major contributor to the overall section length is the large number of different techniques used in the work we present.

6. There are some minor errors/mistakes in manuscript and figures. Please re-check them very carefully and thoroughly. Some identified was shown below.

- In Figure 1, panel C and D are missing. Figure 1 has been revised to include more panels. Panel labeling is now correct.

- In Figure3, put “WT” on the top of Panel A,C,E column and put “P53-/-” on the top of Panel B,D,F column. Suggestion taken and changes made.

- In Figure4C, the cell number between groups is too different. Show the representative images which have similar cell number. The difference in the number of cells is due to the different cellular growth rates of the different strains. We felt it was more important to keep the seeding density, growth time, and magnification unchanged between the samples. 

- In Figure5A and B, Put “mitotracker” and “JC-1” on the top of panel A and B for better understanding. Suggestion taken and changes made.

- In Figure5C, put the full name of MTG to mitotracker green for easy understanding. Suggestion taken and changes made.

- In Figure6 and manuscript, the authors used terms “baseline” and “stressed”. It might be better to change “con” and “Dox”. Baseline and stressed have different meanings than control and DOX. In this experiment, cells were exposed to DOX (or not) and during assessment mitochondrial stressors were added. Baseline and stressed refer to before and after the addition of mitochondrial stressors.

- In Figure9B and C, it will be better to move to supporting data. Suggestion taken and this figure is now supplemental Figure S1.

- In Figure 10A, it will be better to move to supporting data. Suggestion taken and the figure is now supplemental Figure S2.

- In Figure 10C, graph (mitochondrial) is missing. This data is now included supplemental data due to the fact that the results were observed and not quantified.

- In Figure11, indications for A and B are missing. Panels are now labeled.

- Page 9, paraformaldehyde, put the % of paraformaldehyde (might be 4%). The % formaldehyde was added to the methods section.

- Page11, what is the pore size of Boyden chamber? The pore size was added to the methods sections.

- Page 15 line11, what is the dose of DOX? All figure legends were reviewed to ensure dosing information is included.

---

## [Decision Letter · Decision Letter 1]

29 Jun 2020

PONE-D-20-07599R1

Doxorubicin-induced p53 Interferes with Mitophagy in Cardiac Fibroblasts

PLOS ONE

Dear Dr. Aune,

Thank you for submitting your manuscript to PLOS ONE. After careful consideration, we feel that it has merit but does not fully meet PLOS ONE’s publication criteria as it currently stands. Therefore, we invite you to submit a revised version of the manuscript that addresses the points raised during the review process.

Your manuscript was reviewed by the same reviewers and one reviewer raised few minor points. Please address those comments.

We look forward to receiving your revised manuscript.

Kind regards,

Partha Mukhopadhyay, Ph.D.

Academic Editor

PLOS ONE

Reviewers' comments:

Reviewer's Responses to Questions

**Comments to the Author**

1. If the authors have adequately addressed your comments raised in a previous round of review and you feel that this manuscript is now acceptable for publication, you may indicate that here to bypass the “Comments to the Author” section, enter your conflict of interest statement in the “Confidential to Editor” section, and submit your "Accept" recommendation.

Reviewer #1: (No Response)

Reviewer #2: All comments have been addressed

2. Is the manuscript technically sound, and do the data support the conclusions?

Reviewer #1: Yes

Reviewer #2: (No Response)

3. Has the statistical analysis been performed appropriately and rigorously? 

Reviewer #1: Yes

Reviewer #2: (No Response)

4. Have the authors made all data underlying the findings in their manuscript fully available?

Reviewer #1: Yes

Reviewer #2: (No Response)

5. Is the manuscript presented in an intelligible fashion and written in standard English?

Reviewer #1: Yes

Reviewer #2: (No Response)

6. Review Comments to the Author

Reviewer #1: The present version of "Doxorubicin-induced p53 Interferes with Mitophagy in Cardiac Fibroblasts" made improvement compared with the previous version. Here are some minor concerns to address:

1. The authors made the main conclusion that p53 hindering the Parkin localization to mitochondria is the culprit of insufficient mitophagy and mitochondrial dysfunction. However, there is no direct evidence to the mitophagy of p53 deficient cells because result of Figure 4E failed. The authors should perform the same experiment in Figure S1 in p53 deficient cells and compare with the WT cells.

2. According to Figure 6 and Figure S2, Parkin binds to p53 so that it cannot localize to the mitochondria and induce mitophagy after DOX treatment. Why would the staining of Figure 6 show a granule pattern? Are they on mitochondria?

3. In figure 1ABC the WT and p53 deficient cells both showed suppressed growth treated with DOX. However, in figure 1GJ, the cell cycle pattern is much better for p53 deficient cells. The authors need to provide some explanation why the p53 deficient cells do not show much better growth than WT cells under DOX treatment.

4. In figure 2A, the p53 deficient cells have similar pattern of ROS production. P53 deficient cells has higher ROS than WT cells. The following data all address the better mitochondrial function in p53 cells. The authors need to discuss the possible source of ROS and the reason p53 deficient cells do not get more damage.

5. If the bizarre result of Figure 4E is because of their difficulty to collect reliable data from p53 deficient cells, please delete the result. It is hard to believe every single mitochondria is going through mitophagy in p53 deficient cells in any condition. Also, Parkin abundancy is not sufficient to explain the authors’ difficulty getting the data. Parkin is not directly related to the method in this experiment. The authors need to thoroughly discuss why they cannot get the exact percentage of mitochondria going through mitophagy in p53 deficient cells.

6. P53 deficiency was introduced by deleting exons 2-6, which includes the translation starting site, not the promoter. The authors need to correct the description of this gene modulation.

7. In Figure S2D, mitochondrial Parkin western blot should be statistically analyzed. It is also hard to see a molecular weight shift up. Also, if FCCP exerts its action without the involvement of p53, why would we not see a similar band of Parkin in the p53 deficient mitochondria in the FCCP group?

8. The authors need to describe Figure 5B in the results part. Why is p53 found in these cells? What is the pattern of Parkin staining?

Reviewer #2: (No Response)

7. PLOS authors have the option to publish the peer review history of their article (what does this mean?). If published, this will include your full peer review and any attached files.

Reviewer #1: No

Reviewer #2: No

---

## [Author Response · Author response to Decision Letter 1]

13 Aug 2020

Dear Reviewers:

Thank you for the thoughtful and clear critiques of our resubmitted manuscript. Below we have responded to each of the points raised. For clarity, each review point is included in black text and our response in blue text.

1. The authors made the main conclusion that p53 hindering the Parkin localization to mitochondria is the culprit of insufficient mitophagy and mitochondrial dysfunction. However, there is no direct evidence to the mitophagy of p53 deficient cells because result of Figure 4E failed. The authors should perform the same experiment in Figure S1 in p53 deficient cells and compare with the WT cells. 

To provide further clarification we would respectfully point out that the results did not fail. Gating for positive/negative lysosomal cells and positive/negative mitophagy cells was done under the supervision of the UTHSCSA Flow Cytometry Core using WT samples (control, DOX, FCCP) without dyes and with each dye separately. The intensity of each dye was maximally increased. The increased intensity in the mitophagy dye alone demonstrates the increase in the process. It is the acidification/intensification of the mitophagy dye that demonstrates the fusion of the phagosome with the lysosome. Due to the excess amount of Parkin at baseline, the majority of the p53-/- cells in each sample group were undergoing maximal mitophagy. Overall, this demonstrates that in the absence of p53, these cells are able to undergo mitophagy. The mitophagy assay was repeated in the p53-/- cells and visualized with fluorescent microscopy. In our revision, this experimental data was added as panel B in Fig S1. The middle panel of Fig S1B is an example of the mitophagy dye in the p53-/- cells. Compared to the control cells of the WT cells, a significant increase in the amount and intensity of the red fluorescence can be seen in every treatment group of the p53-/- cells. The fluorescence of the green lysosome dye is also increased in each treatment group.

2. According to Figure 6 and Figure S2, Parkin binds to p53 so that it cannot localize to the mitochondria and induce mitophagy after DOX treatment. Why would the staining of Figure 6 show a granule pattern? Are they on mitochondria? 

Figure 6B and 6C are not indicative of a granular staining pattern per se, but rather the appearance related to the methodology used to visualize the interaction. The fluorescence arises from the proximity of complimentary DNA probes that are ligated together and then amplified with fluorescent oligonucleotides. There are multiple examples from the literature that demonstrate the fluorescent pattern with the proximity ligation assay. However, considering the proximity that transcription factors like p53 and Parkin have to the nucleus it would not be surprising that the interactions of the two proteins would be near the nucleus and/or mitochondria. Moreover, Parkin has the additional function of regulating mitophagy via its translocation between the mitochondria and cell nucleus. Both the results and discussion have been revised to further clarify these considerations.

Support from the literature (references added to manuscript)

• Alves de Costa: https://www.ncbi.nlm.nih.gov/pmc/articles/PMC6341214/

• Al-Mehdi: https://www.ncbi.nlm.nih.gov/pmc/articles/PMC3565837/

• Park: https://www.ncbi.nlm.nih.gov/pmc/articles/PMC125431/

3. In figure 1ABC the WT and p53 deficient cells both showed suppressed growth treated with DOX. However, in figure 1GJ, the cell cycle pattern is much better for p53 deficient cells. The authors need to provide some explanation why the p53 deficient cells do not show much better growth than WT cells under DOX treatment. The decreased proliferation seen in the p53-/- is likely due to a combination of things. DOX alters gene expression. One study demonstrated repression of pathways normally activated in the G1 phase of the cell cycle. These pathways are integral to the “doubling” of cell contents necessary for proliferation. Inhibiting mitophagy can often lead to apoptosis, likely via cell stress pathways including p53-mediated cell cycle arrest. However, inducing mitophagy has been shown to decrease cell proliferation simply due to the high energy demand of mitophagy and mitochondrial biogenesis.

Support from the literature (references added to manuscript)

• Nikerel: https://www.ncbi.nlm.nih.gov/pmc/articles/PMC6135803/

• Boyle: https://www.jbc.org/content/293/38/14891.short

• Pang: https://www.ncbi.nlm.nih.gov/pmc/articles/PMC3674280/

• Nanasi: https://journals.plos.org/plosone/article?id=10.1371/journal.pone.0231223

• Kang: https://www.tandfonline.com/doi/full/10.1080/21691401.2019.1593854

• Wei: https://link.springer.com/article/10.1007/s12192-018-0937-7

• Niu: https://www.sciencedirect.com/science/article/pii/S1382668916301442

4. In figure 2A, the p53 deficient cells have similar pattern of ROS production. p53 deficient cells has higher ROS than WT cells. The following data all address the better mitochondrial function in p53 cells. The authors need to discuss the possible source of ROS and the reason p53 deficient cells do not get more damage. 

Figure 2A addresses ROS production in the entire cell, not just at the mitochondria. Figure 2B shows that there is a greater variability in the production of ROS associated with the mitochondria in p53 null cells, whereas mitochondrial ROS is more uniformly increased in WT cells. The variability of mitochondrial ROS production of p53 null cells could reflect the process of mitophagy, as dysfunctional ROS-producing mitochondria are degraded. Additionally, both groups are relative to their control baseline. Since it appears that mitophagy is increased at baseline in the p53 null cells, the baseline mitochondrial ROS production could be decreased as mitochondria are more efficiently cleared. DOX has multiple mechanisms of producing ROS in cells. Some of these are mediated via mitochondria and mitochondrial dysfunction. However, DOX can create free radicals by interacting with Fe2+/3+ in the cell. The cellular increase of ROS seen in WT and p53 null cells is a combination of DOX/Fe and mitochondrial production. The scope of our paper is limited to the ROS produced by mitochondria and the resulting mitochondrial dysfunction. Other organelles of the cell are also damaged by the increase in cellular ROS and we did not study the effect of p53 on these other organelles. It is likely that there are many significant changes in the p53 null cells that are beneficial to the other organelles.

Support from the literature (references added to manuscript)

• DOX ROS production: https://clincancerres.aacrjournals.org/content/20/18/4737 ; https://www.sciencedirect.com/science/article/pii/S0167488916300118

• Nuclear damage: https://www.pnas.org/content/117/26/15182

• Golgi damage: https://www.ncbi.nlm.nih.gov/pmc/articles/PMC3374628/

• ER damage: https://pubmed.ncbi.nlm.nih.gov/31580970/ ; https://pubmed.ncbi.nlm.nih.gov/26838784/

• Membrane damage: https://pubmed.ncbi.nlm.nih.gov/29019935/ ; https://journals.plos.org/plosone/article?id=10.1371/journal.pone.0026441

5. If the bizarre result of Figure 4E is because of their difficulty to collect reliable data from p53 deficient cells, please delete the result. It is hard to believe every single mitochondria is going through mitophagy in p53 deficient cells in any condition. Also, Parkin abundancy is not sufficient to explain the authors’ difficulty getting the data. Parkin is not directly related to the method in this experiment. The authors need to thoroughly discuss why they cannot get the exact percentage of mitochondria going through mitophagy in p53 deficient cells. 

The discrepancy comes from measuring mitophagy in cells and not individual mitochondria. WT samples were used to define the population of Mito+, Lyso+, and “double positive” cells based on the amount of fluorescence. 98-100% of the cells from the p53 null samples showed max fluorescence with the lysosome dye and the mitophagy dye. Additional comments on this population have been added to results section.

6. p53 deficiency was introduced by deleting exons 2-6, which includes the translation starting site, not the promoter. The authors need to correct the description of this gene modulation. 

Corrected in the manuscript

7. In Figure S2D, mitochondrial Parkin western blot should be statistically analyzed. It is also hard to see a molecular weight shift up. Also, if FCCP exerts its action without the involvement of p53, why would we not see a similar band of Parkin in the p53 deficient mitochondria in the FCCP group? 

This figure was moved to the supplemental section at the suggestion of the reviewers, as opposed to removing it from the paper completely. The MW shift up can be seen in S2C where the lighter Parkin bands can be seen in each treatment group in the mitochondrial fraction. It is a small MW shift, indicative of an addition of only 5-10 kDa. There should be a Parkin band in FCCP-treated sample in the mitochondrial fraction of the p53 null cells. This band was seen in other blots, but the band was not seen in the DOX-treated cells. We believe this is due to the instability of the modified protein. Samples in which the band was previously seen would be rerun on the same blot for quantification and imaging, but not appear this time. 

8. The authors need to describe Figure 5B in the results part. Why is p53 found in these cells? What is the pattern of Parkin staining? 

The diffuse, faint staining seen in figure 5B is not specific to p53. The puncta of the fluorescence does not have the same intensity as that seen in the WT examples. This diffuse patterning also appears to “shade” in the shape of the cell. This background fluorescence is also slightly visible in the WT cells. The contrast of the images were adjusted together so that the p53 staining could be seen when merged with the Parkin images. This adjustment in contrast enabled in the visualization of the background fluorescence. Figure 5B is addressed more specifically in the text now.

---

## [Editor Report · Decision Letter 2]

26 Aug 2020

Doxorubicin-induced p53 Interferes with Mitophagy in Cardiac Fibroblasts

PONE-D-20-07599R2

Dear Dr. Aune,

We’re pleased to inform you that your manuscript has been judged scientifically suitable for publication and will be formally accepted for publication once it meets all outstanding technical requirements.

Kind regards,

Partha Mukhopadhyay, Ph.D.

Section Editor

PLOS ONE
---

## [Editor Report · Acceptance letter]

11 Sep 2020

PONE-D-20-07599R2 

Doxorubicin-induced p53 Interferes with Mitophagy in Cardiac Fibroblasts 

Dear Dr. Aune:

I'm pleased to inform you that your manuscript has been deemed suitable for publication in PLOS ONE. Congratulations! Your manuscript is now with our production department. 

Kind regards, 

on behalf of

Dr. Partha Mukhopadhyay 

Section Editor

PLOS ONE